# Semen inhibits Zika virus infection of cells and tissues from the anogenital region

Janis A. Müller[1], Mirja Harms[1], Franziska Krüger[1], Rüdiger Groß [1], Simone Joas[1], Manuel Hayn[1], Andrea N. Dietz[2], Sina Lippold [2], Jens von Einem[2], Axel Schubert[2], Manuela Michel[2], Benjamin Mayer[3], Mirko Cortese[4], Karen S. Jang[5,6], Nathallie Sandi-Monroy[1], Miriam Deniz[7], Florian Ebner[7,8], Olli Vapalahti[9,10], Markus Otto[11], Ralf Bartenschlager[4,12], Jean-Philippe Herbeuval[13], Jonas Schmidt-Chanasit[14,15], Nadia R. Roan[5,6] & Jan Münch[1,16]

Zika virus (ZIKV) causes severe birth defects and can be transmitted via sexual intercourse. Semen from ZIKV-infected individuals contains high viral loads and may therefore serve as an important vector for virus transmission. Here we analyze the effect of semen on ZIKV infection of cells and tissues derived from the anogenital region. ZIKV replicates in all analyzed cell lines, primary cells, and endometrial or vaginal tissues. However, in the presence of semen, infection by ZIKV and other flaviviruses is potently inhibited. We show that semen prevents ZIKV attachment to target cells, and that an extracellular vesicle preparation from semen is responsible for this anti-ZIKV activity. Our findings suggest that ZIKV transmission is limited by semen. As such, semen appears to serve as a protector against sexual ZIKV transmission, despite the availability of highly susceptible cells in the anogenital tract and high viral loads in this bodily fluid.

[1] Institute of Molecular Virology, Ulm University Medical Center, 89081 Ulm, Germany. [2] Institute of Virology, Ulm University Medical Center, 89081 Ulm, Germany. [3] Institute of Epidemiology and Medical Biometry, Ulm University, 89075 Ulm, Germany. [4] Department of Infectious Diseases, Molecular Virology, Medical Faculty, Heidelberg University, 69120 Heidelberg, Germany. [5] Gladstone Institute of Virology and Immunology, San Francisco, CA 94158, USA. [6] Department of Urology, University of California, San Francisco, San Francisco, CA 94158, USA. [7] Klinik für Frauenheilkunde und Geburtshilfe, Ulm University Medical Center, 89081 Ulm, Germany. [8] Frauenklinik, Helios Amper Klinik, 85221 Dachau, Germany. [9] Department of Virology and Immunology, University of Helsinki and Helsinki University Hospital, 00014 Helsinki, Finland. [10] Department of Veterinary Biosciences, University of Helsinki, 00014 Helsinki, Finland. [11] Department of Neurology, Ulm University, 89081 Ulm, Germany. [12] German Center for Infection Research (DZIF), Heidelberg Partner Site, Heidelberg University, 69120 Heidelberg, Germany. [13] Chemistry, Biology, Modeling and Immunotherapy (CBMIT), CNRS, UMR8601, Laboratoire de Chimie et Biochimie Pharmacologiques et Toxicologiques, Université Paris Descartes, CICB Paris, 75006 Paris, France. [14] Bernhard Nocht Institute for Tropical Medicine, World Health Organization Collaborating Centre for Arbovirus and Hemorrhagic Fever Reference and Research, 20359 Hamburg, Germany. [15] German Centre for Infection Research (DZIF), Partner Site Hamburg-Luebeck-Borstel, 20359 Hamburg, Germany. [16] Core Facility Functional Peptidomics, Ulm University Medical Center, 89081 Ulm, Germany. Correspondence and requests for materials should be addressed to J.M. (email: jan.muench@uni-ulm.de)

Zika virus (ZIKV) was first identified in 1947 in Uganda[1] but was not thought to be a significant threat to humans. When the virus re-emerged in 2007, however, it rapidly caused a series of epidemics in Micronesia[2], the South Pacific[3], and the Americas[4]. In March 2017, 64 countries and territories reported ongoing viral transmission[5], primarily through some species of mosquitoes. ZIKV can cause numerous diseases in adults, including meningoencephalitis[6], myelitis[7], and Guillain-Barré syndrome[8]. Of even greater concern is the observation that the ZIKV epidemics were associated with a dramatic increase in cases of microcephaly in newborns. Several studies revealed that ZIKV infection during pregnancy can directly cause fetal demise, microcephaly, and other congenital problems[9] and that disease may develop in up to 46% of the cases[10].

Atypical for an arthropod-borne virus, ZIKV has also been reported to be transmitted via sexual intercourse (reviewed by refs. [11–15]). Up to now, 13 countries reported several cases of sexual transmission of ZIKV[5], resulting in classification of this virus as a sexually transmitted pathogen. These case reports describe sexual ZIKV transmission not only by symptomatic but also by asymptomatic individuals. Semen (SE) from an infected individual can harbor ZIKV at extremely high concentrations of up to $10^8$ viral RNA copies per ml[16–20] which are 4–5 log-fold higher than that present in serum, urine, and saliva, and the virus can remain detectable in SE >6 months after onset of symptoms[20–23]. These observations have led to the notion that ZIKV in SE may be responsible for many cases of viral transmission[16,23–28]. As a result, the World Health Organization has advised males and females with confirmed Zika fever or those who have traveled to areas with active ZIKV circulation to consider using condoms or staying abstinent for a period of at least 6 months[29]. However, the true contribution of sexual transmission to the epidemic spread of ZIKV is currently unclear[30]. A recent study estimated the overall sexual ZIKV transmission rate to be as high as 3%[31]. This number, however, is based on mathematical models rather than clinical and epidemiological data. Whether ZIKV efficiently transmits sexually is of high importance, because if so, this sexually transmitted disease may contribute significantly to microcephaly cases. Consistent with the idea that ZIKV can be sexually transmitted are in vivo experiments suggesting that vaginal ZIKV exposure of pregnant mice or mating of ZIKV-infected mice with naive females results in viral transmission and infection of the fetus[32–34] where it causes malformations or fetal demise[35].

During most cases of sexual ZIKV transmission, SE is the viral vehicle. SE is rich in bioactive organic and inorganic substances, including proteins, enzymes, polyamines, cytokines, chemokines, hormones, and ions[36]. These soluble components can induce transient changes in the vaginal milieu that may influence the efficiency of virus transmission[37]. In the case of human immunodeficiency virus-1 (HIV-1), a virus that is predominantly transmitted sexually, SE markedly enhances its infectivity[38–41]. The HIV-enhancing activity of SE has been attributed to amyloid fibrils naturally present in SE. These fibrils form by self-assembly of peptide fragments derived from the seminal proteins prostatic acid phosphatase (PAP) and semenogelins[38–40,42–44]. Seminal amyloid has a positive surface charge that allows it to bind to and concentrate the negatively charged HIV particles, thereby increasing their attachment to and viral entry into cellular targets[42,43,45]. The best-characterized SE amyloid forms from the PAP248-286 peptide and is termed semen-derived enhancer of virus infection (SEVI)[38]. Several compounds that counteract the infection-promoting activity of seminal amyloid have been described and are being considered as leads for microbicide development[46,47]. The effect of SE and seminal amyloids on ZIKV infection has not yet been addressed[35,48] but could

contribute to a better understanding of ZIKV as a sexually transmitted disease[30,49].

Since ZIKV is a sexually transmittable pathogen, we studied whether it replicates in anogenital tissues and whether SE and seminal amyloid affect ZIKV infection. We show that ZIKV efficiently replicates in cells and tissues derived from the anogenital region. Surprisingly, seminal amyloid does not affect ZIKV infection, while SE markedly suppresses ZIKV infection through blocking viral attachment to target cells. These results can help explain the low frequencies of ZIKV transmission by sexual intercourse despite the high viral loads detectable in SE and the high susceptibility of anogenital cells to ZIKV infection.

## Results

**ZIKV infects and replicates in cells of the anogenital tract.** As a sexually transmitted virus, ZIKV needs to productively infect cells present at mucosal portals of entry, e.g., the female reproductive tract (FRT) or the rectum. To clarify whether cells of anogenital origin support productive infection, primary endometrial stromal fibroblasts (eSFs) and human foreskin fibroblasts (HFFs), as well as cell lines derived from endometrium (HeLa, TZM-bl), colon (SW480 and T-84), or ovaries (OVCAR-3 and SKOV3), were inoculated with African ZIKV isolate MR766[1]. Two hours postinfection, cells were washed, fresh medium was added, and 2 days later cells were stained for the viral E protein. Confocal microscopy demonstrated that all analyzed cell types were infected as assessed by E protein expression (Fig. 1a). All infected cells released viral RNA (Fig. 1b) and infectious virus (Fig. 1c). Among the tested cell lines, viral replication was least efficient in SKOV3 and T-84 cells and most efficient in SW480, HeLa, and OVCAR-3 cells. Primary endometrial and foreskin fibroblasts also both supported efficient levels of ZIKV replication. These results demonstrate that ZIKV is capable of establishing productive infection in cells derived from the anogenital region.

**ZIKV replicates in ex vivo endometrial and vaginal tissues.** To analyze whether ZIKV also replicates in intact tissues isolated from the FRT, surgically removed vaginal (VT) or endometrial tissue (ET) was cut into tissue blocks and infected with ZIKV, as previously described for HIV-1 infection experiments[50–53]. For these experiments, we used ZIKV MR766 and the recent pandemic FB-GWUH-2016 (GWUH) isolate derived from the brain of an aborted fetus[54]. After 4 h, tissues were extensively washed to remove viral inoculum, and six tissue blocks were placed at the air/liquid interface on gel-foams in 12-well plates. Productive virus infection was assessed by quantifying infectious progeny virus in supernatants collected from day 0 (wash control) up to day 8 postinfection. Titers that increased more than 10-fold as compared to the wash control were considered indicative of productive infection. Both ZIKV strains established productive infection in the majority of the analyzed VTs and ETs (Fig. 2 and Table 1). The absolute titers and kinetics of replication, however, varied between experiments and donors. For example, ZIKV GWUH replicated in 6 out of the 7 VTs (frequency of 86%) with a maximum titer of $2.32 \times 10^6$ tissue culture infectious dose 50 ($TCID_{50}$)/ml observed at day 8 postinfection for tissue derived from donor #1 (Fig. 2 and Table 1). ZIKV MR766 established infection in 66% of VTs (4 out of the 6 tissues) (Fig. 2a and Table 1) with the highest titer observed at day 4 in tissue from donor #6. In all seven ET samples analyzed, ZIKV replication was observed (Fig. 2c, d, Table 1), with the highest titer being $1.6 \times 10^6$ $TCID_{50}$/ml at day 8 with tissue from donor #13.

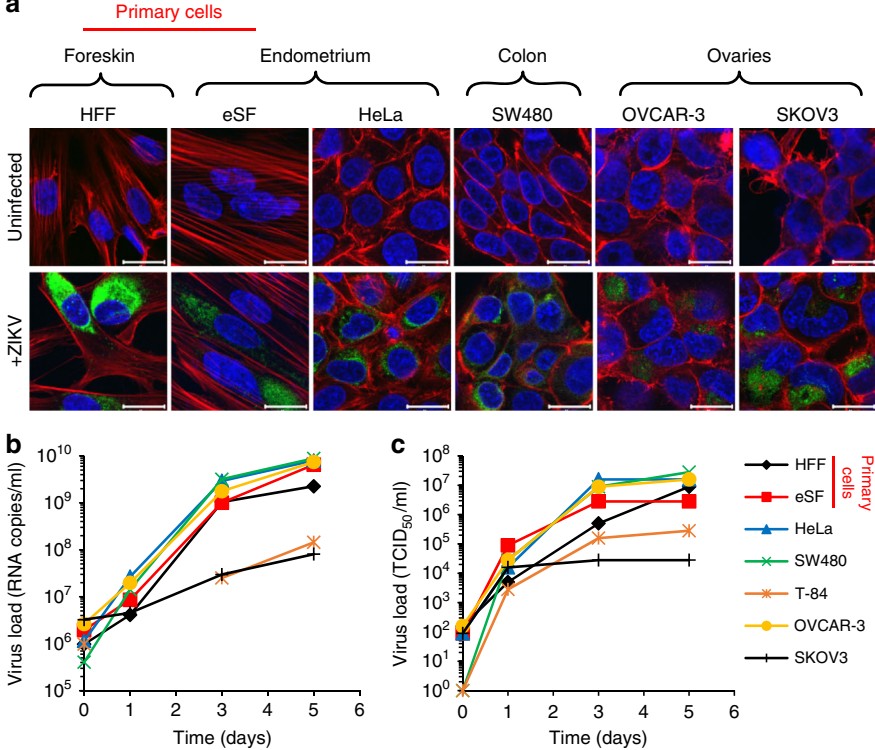

**Fig. 1** ZIKV infects and replicates in cells of the anogenital tract. **a** Primary HFFs and eSFs and cell lines derived from the cervix (HeLa), colon (SW480), or ovaries (OVCAR-3 and SKOV3) were inoculated with ZIKV MR766. Two days later, cells were stained for flavivirus protein E (green), nuclear DNA (blue), and actin (red) and imaged by confocal microscopy. Scale bars correspond to 20 μm. **b** Supernatants of infected cells of the anogenital tract were taken at days 0, 1, 3, and 5, and viral genome copy numbers were determined by qPCR. **c** Infectious virus titer of supernatants was determined by TCID$_{50}$ titration onto Vero E6 cells. HFF: human foreskin fibroblast, eSF: endometrial stromal fibroblast

To further determine the variability of viral replication in an individual tissue, a vaginal sample was cut into 48 blocks, and all blocks were infected in the same tube but analyzed individually in individual wells of a microtiter plate. As shown in Supplementary Fig. 1, ZIKV replicated in 66% (32 out of 48) of the blocks. Thus, although there was considerable variability in these assays, the results unequivocally demonstrate that ZIKV replicates ex vivo in VTs and ETs.

**Seminal amyloid does not alter ZIKV infection rates**. As it has previously been reported that amyloid fibrils in SE boost HIV-1 infection[38–40,42–44], we were interested in whether these fibrils may exert similar effects on ZIKV. To test this, the two ZIKV strains were exposed to physiologically relevant concentrations of SEVI fibrils[38]. After 5 min, SEVI-treated viruses were added to Vero E6 cells, which are highly permissive for ZIKV[55]. Infection rates were determined 2 days later by a cell-based ZIKV immunodetection assay that quantifies the flavivirus protein E[56]. Results shown in Supplementary Fig. 2a demonstrate that SEVI fibrils had no effect on infection by either ZIKV strain. To ensure that the SEVI fibrils were active, we next measured the effect of SEVI on HIV-1 and ZIKV infection of TZM-bl cells, an HeLa-derived cell line permissive for HIV-1[57] and ZIKV (Fig. 1). ZIKV infection rates were determined as described above, while HIV-1 infection rates were monitored by the expression of viral p24 capsid antigen instead of protein E from ZIKV. While HIV-1 infection was markedly increased by SEVI, ZIKV infection was unaffected (Supplementary Fig. 2b). Thus, seminal amyloid does not enhance the infectivity of ZIKV, in contrast to other sexually transmitted viruses like HIV-1[38], herpes simplex virus-2 (HSV-2)[58], and cytomegalovirus (CMV)[59].

**SE and seminal plasma inhibit ZIKV infection**. Given the importance of SE fibrils for the HIV-enhancing effect of SE[38–40,43], and our observation that these fibrils do not enhance ZIKV infection (Supplementary Fig. 2), we anticipated that SE, like SE fibrils, would not increase ZIKV infection. To analyze the effects of SE on ZIKV infection, ejaculates from 20 individuals were liquefied and then combined to generate pooled SE. The pooled SE was then aliquoted and frozen at −80 °C or centrifuged to obtain the cell-free supernatant, termed seminal plasma (SP), which was also aliquoted and frozen at −80 °C. SE and SP were titrated onto Vero E6 cells at concentrations up to 5%, and then cells were infected with both ZIKV strains. To minimize cytotoxic effects of SE and SP[39,60,61], the inoculum was removed after 2 h, and fresh medium without SE/SP was added. Infection rates were determined 2 days later by quantifying ZIKV E protein in a cell-based immunodetection assay. Surprisingly, SE and SP effectively suppressed ZIKV infection (Fig. 3a). A final cell culture concentration of 1% SE or SP reduced ZIKV MR766 and GWUH infection by >90%, and a concentration of 5% SE or SP almost entirely abrogated infection (Fig. 3a), in the absence of any cytotoxicity (Supplementary Fig. 3). Effective inhibition of ZIKV infection by SP and SE was confirmed by fluorescence microscopy (Fig. 3b–d and Supplementary Fig. 4) and flow cytometry (Fig. 3e and Supplementary Fig. 5). To exclude the possibility that SE or SP may interfere with antibody-based detection of the viral antigen, cells that were infected in the absence or presence of SE or SP were also analyzed microscopically. At day 5 postinfection, ZIKV MR766 and GWUH induced massive cytopathic effects (CPEs), as indicated by rounded and detached cells (Supplementary Fig. 6). In the presence of 1% SE or SP, no CPE was visible in ZIKV-exposed cells, and cells were morphologically similar to uninfected cells (Supplementary Fig. 6). These results are consistent with the

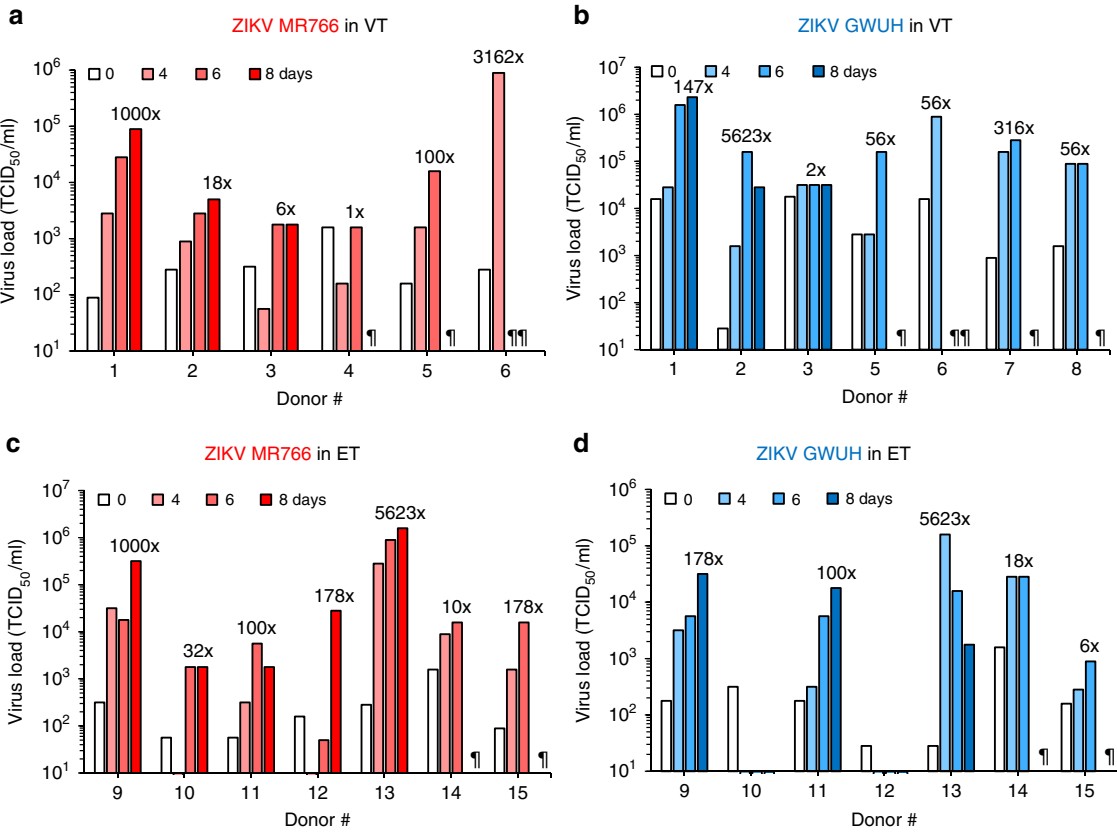

**Fig. 2** ZIKV infects and replicates in primary vaginal and endometrial tissue. **a**, **b** Primary vaginal tissue (VT) or **c**, **d** endometrial tissue (ET) blocks were infected for 2 h with ZIKV MR766 (**a**, **c**, in red) or GWUH (**b**, **d**, in blue). Blocks were then washed in PBS, transferred onto foams, and cultured for up to 8 days. Aliquots of the supernatants were taken at day 0 (wash control) and days 4, 6, and 8 and then analyzed for infectious titer by $TCID_{50}$ titration on Vero E6 cells. Maximum fold-increase titers were compared to the wash control (day 0) and are indicated in italics above the corresponding bars. ¶No samples collected on that day. See also Table 1

| Table 1 Replication of ZIKV in ex vivo-infected vaginal and endometrial tissues | | | | |
|---|---|---|---|---|
| | **ZIKV MR766** | | **ZIKV GWUH** | |
| | VT (*n* = 6) | ET (*n* = 7) | VT (*n* = 7) | ET (*n* = 7) |
| Productive infection[a] | 4/6 | 7/7 | 6/7 | 4/7 |
| Max. titer ($TCID_{50}$/ml)[b] | $8.89 \times 10^5$ (#6, day 4) | $1.58 \times 10^6$ (#13, day 8) | $2.32 \times 10^6$ (#1, day 8) | $1.58 \times 10^5$ (#13, day 4) |

VT vaginal tissue, ET endometrial tissue, # donor
[a] Number of tissues that were productively infected/number of all tissues analyzed; tissues were considered to be productively infected if infectious virus titers were ≥10-fold greater than the wash control
[b] Highest ZIKV titer measured; the numbers in brackets indicate the donor and the day postinfection that the supernatant was taken

inability of SE fibrils to enhance ZIKV infection and further show that human SE potently inhibits ZIKV infection.

**ZIKV suppression is a general property of human SE.** As the previous set of experiments used pooled SE/SP, it was possible that a potent inhibitor of ZIKV infection was present in SE from only a subset of individuals. To assess how common this inhibitor is, we next determined the inhibitory activity of ten ejaculates from different donors. All ten ejaculates efficiently inhibited ZIKV GWUH (Fig. 4a) and MR766 (Supplementary Fig. 7a) infection. The average half-maximal inhibitory concentration ($IC_{50}$ ± standard deviation of 3 replicates) was 0.74 ± 0.18% for ZIKV GWUH, and 0.84 ± 0.11% for MR766 (Table 2). Again, 5% SE almost completely blocked infection of both ZIKV strains. Similar results were obtained with SE that was pooled from these ten donors with $IC_{50}$s against GWUH and MR766 of 0.67 and

0.76%, respectively (Fig. 4a, Supplementary Fig. 7a and Table 2). Thus, the inhibitory activity of SE appears to be a general property and is not donor specific.

**SE inhibits ZIKV by blocking viral attachment to cells.** Results shown in Figs. 3 and 4a and Supplementary Fig. 7a were obtained by exposing cells to 5, 1, 0.2, 0.04, and 0% SE, followed by inoculation with ZIKV ("cell treatment"). To determine whether the ZIKV inhibitory activity is directed against the cell or the viral particle, the two ZIKV strains were first treated with 50, 10, 2, 0.4, and 0% SE and then added to target cells ("virion treatment"). This experimental set-up resulted in a ten-fold higher SE concentration during virion incubation but same final SE concentrations in cell culture as the "cell treatment" set-up. As shown in Fig. 4b and Supplementary Fig. 7b, SE concentrations up to 50% during virus pretreatment efficiently inhibited ZIKV

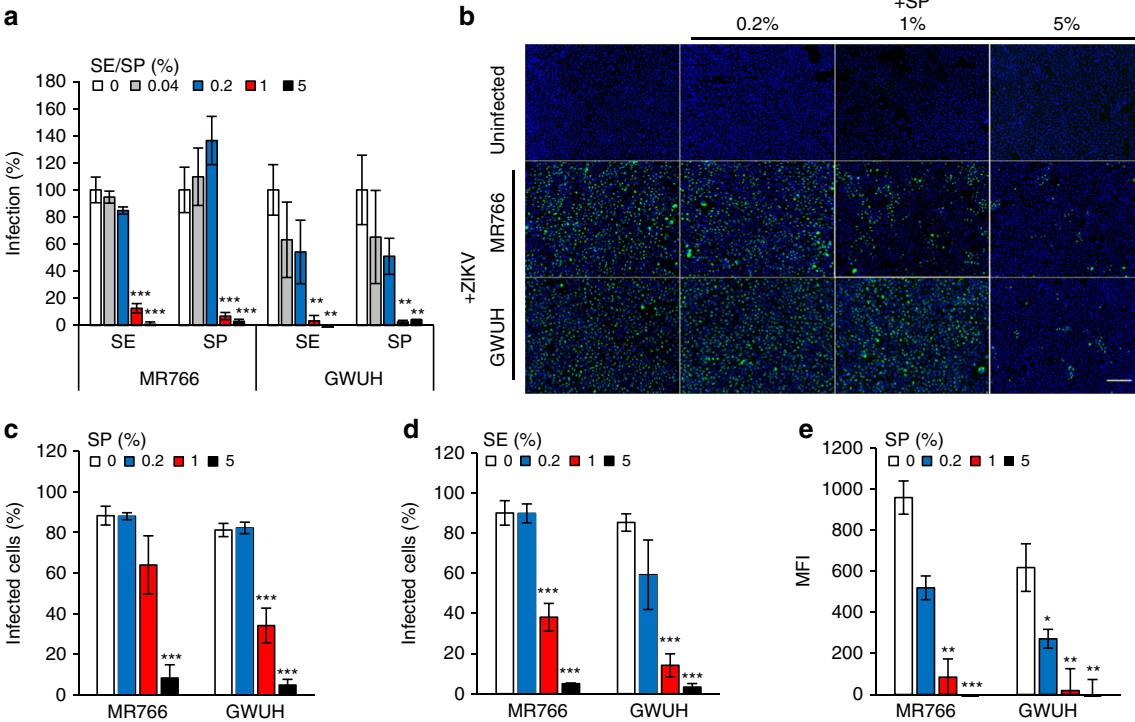

**Fig. 3** Semen and seminal plasma inhibit ZIKV infection. **a** Vero E6 cells were incubated with the indicated concentrations of SE or SP and inoculated with ZIKV MR766 or GWUH. Two days later, infection rates were determined by a cell-based ZIKV immunodetection assay that enzymatically quantifies the flavivirus protein E. Data are normalized to infection rates in absence of SE or SP and represent average values obtained from triplicate infections ± standard deviations. **b** Vero E6 were incubated with the indicated concentrations of SP before inoculation with the two ZIKV strains. Two days postinfection, cells were stained for flavivirus protein E (green) and nuclear DNA (blue) and visualized by fluorescence microscopy. Scale bar corresponds to 200 μm. **c** Quantification of protein E-positive cells in **b** ±standard deviation of triplicate infections, conducted with ImageJ. **d** Quantification of ZIKV infection in presence of SE based on microscopy data (see Supplementary Fig. 4). **e** Mean fluorescence intensity (MFI) of ZIKV-infected Vero E6 cells assessed by flow cytometry (see Supplementary Fig. 5). *$P < 0.01$, **$P < 0.001$, ***$P < 0.0001$ (by one-way ANOVA with Bonferroni post-test)

infection. Of note, the $IC_{50}$s were similar to those obtained for the "cell treatment" set-up (Table 2). These observations demonstrate that final SE concentrations in cell culture, rather than the concentrations acting on the virion, determine the efficiency of inhibition, thereby suggesting that the inhibitory factor is acting on the target cell and not the virion.

To clarify which step in the viral life cycle is blocked, "time of addition" experiments were performed. When ZIKV and SP were added simultaneously, infection was blocked, as expected (Fig. 4c, ZIKV+SP). In contrast, when cells were first infected, followed by a washing step and addition of SP 2 h later (after the virus has already entered the cell), no antiviral effect was observed (Fig. 4c, ZIKV→SP). These data suggest that SP inhibits an early event in the viral life cycle. Alternatively, if cells were first treated with SP and then washed to remove the inhibitor, limited inhibition was observed (Fig. 4c, SP→ZIKV) demonstrating that the antiviral activity of SP is reversible. To clarify whether SP may interfere with ZIKV binding to the target cell, we established a confocal microscopy-based assay that quantifies ZIKV attachment. For this, a confluent layer of Vero E6 cells was inoculated with increasing concentrations of ZIKV for 2 h at 37 °C or 4 °C, then cells were fixed, washed, attached virions were stained, and a z-stack of confocal microscopy images was taken. Cell-associated ZIKV particles were readily detectable (Supplementary Figs. 8a, 9a) and quantifiable (Supplementary Figs. 8b, 9b) at a multiplicity of infection (MOI) >1. Increasing concentrations of SP reduced fluorescence intensities in a dose-dependent manner, with 5% SP inhibiting ZIKV attachment almost completely (Fig. 4d, e and Supplementary Figs. 8c, d and 9c, d). Together these data show that SE prevents the initial binding of Zika virions to target cells.

**SE-derived extracellular vesicle preparations inhibit ZIKV.** To investigate whether the factor(s) in SE that is responsible for the anti-ZIKV activity is a peptide or protein, SP was subjected to Proteinase K digestion (Supplementary Fig. 10a) or treatment with heat (Supplementary Fig. 10b) or acid (Supplementary Fig. 10c) to denature proteins. None of these treatments abolished the antiviral activity of SP demonstrating that the antiviral factor is not a polypeptide. Using lectin affinity chromatography (Concanavalin A columns) to remove glycosylated macromolecules from SP, we found that glycoproteins are also not involved in the anti-ZIKV activity (Supplementary Fig. 10d). We then assessed the size of the responsible factor. SP that was syringe filtered through 0.2 μm pores retained antiviral activity (Fig. 5a). When SP was centrifuged through a 300 kDa filter, the antiviral activity was found in the retentate, whereas the filtrate was inactive (Fig. 5b). Thus the anti-ZIKV factor in cell-free SP has a molecular weight above 300 kDa but passes through 200 nm-diameter pores. SE contains huge amounts of structurally diverse, extracellular vesicles (EVs) with sizes between 50 and 400 nm[62,63]. To analyze the effect of large and small EVs on ZIKV infection, SP was centrifuged through 0.65 μm filters, after which the filtrate was transferred to 0.22-μm centrifugal filters, as described[64]. Large EVs were recovered from the top of the filters by washing, while small EVs were obtained as flow through[64]. Both EV preparations inhibited ZIKV infection with EVs <220 nm being as potent as non-processed SP (Fig. 5c). This sample contained particles with an average diameter of 141.6 ± 3.7 nm (Supplementary Fig. 11a). Such particles were largely absent in the 300 kDa filtrate that does not contain inhibitory activity (Fig. 5b and Supplementary Fig. 11b). The SP-derived EV

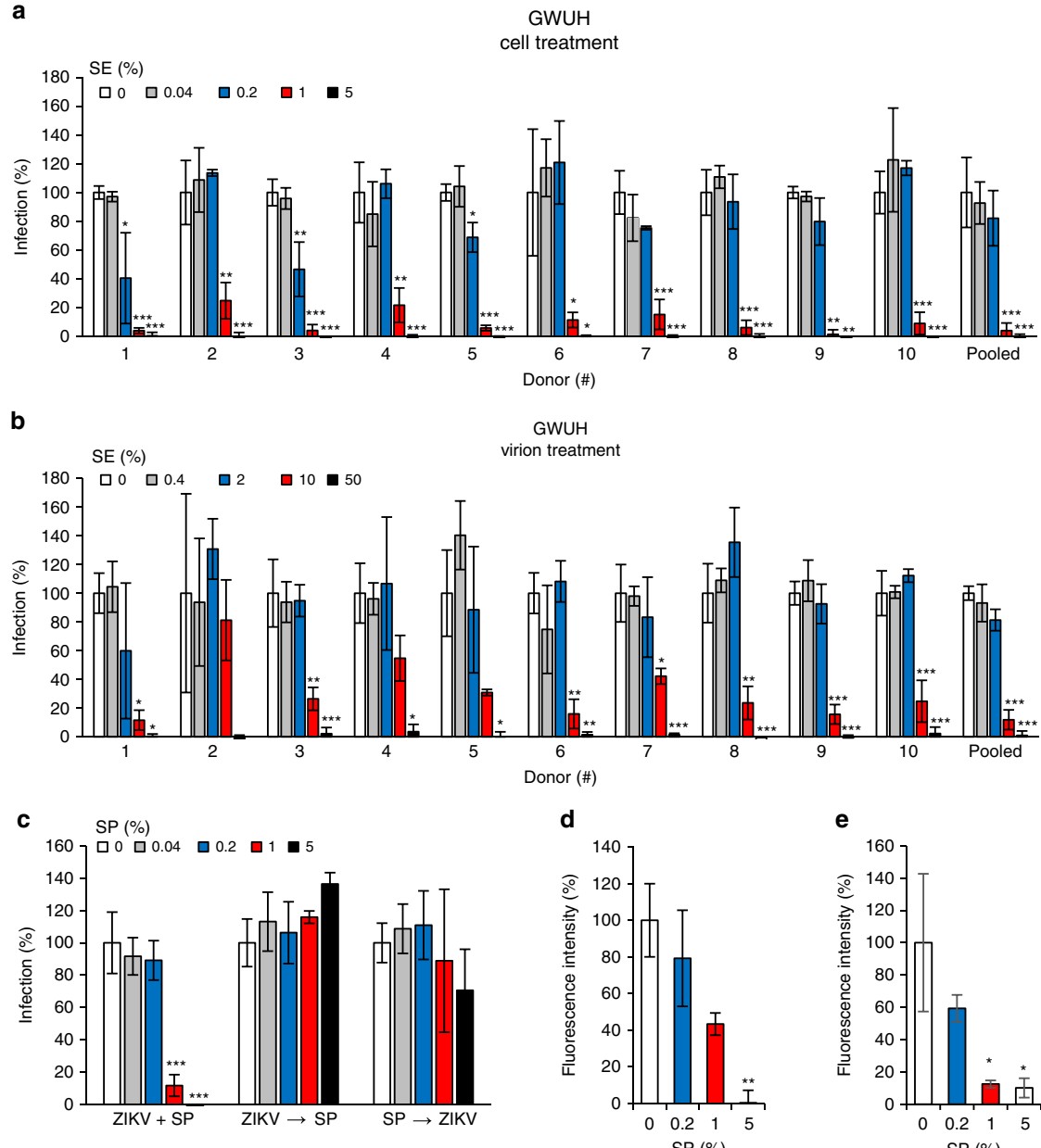

**Fig. 4** The anti-ZIKV activity in semen is donor independent and targets an early step in the viral life cycle. **a** Vero E6 cells were incubated for 10 min with the indicated concentrations of SE derived from ten individual donors or a pool of all ten samples. Cells were then inoculated with ZIKV GWUH and monitored for infection rates. **b** ZIKV was first incubated with 0, 0.4, 2, 10, or 50% of the individual and pooled SE samples for 10 min ("virion treatment") and then added to Vero E6 cells resulting in similar final SE concentrations as shown in **a**. This experimental set-up differs from the "cell treatment" protocol used in **a** in that ZIKV is first treated with high concentrations of SE. Infection rates were determined by a cell-based ZIKV immunodetection assay 2 days postinfection. Data are normalized to corresponding infection in absence of SE/SP and represent average values obtained from triplicate infections ± standard deviations. For IC$_{50}$ values, see Table 2. **c** Vero E6 cells were inoculated with (i) SP and ZIKV simultaneously (ZIKV+SP); (ii) first ZIKV and then, after a washing step, SP 2 h later (ZIKV→SP), or (iii) first SP and then, after a washing step, ZIKV 2 h later (SP→ZIKV). After another 2 h, medium was changed and 2 days later infection rates measured. **d** ZIKV virions were allowed to attach to Vero E6 cells in the presence of the indicated concentrations of SP for 2 h at 37 °C or **e** at 4 °C. Cells were then washed and stained for ZIKV protein E and cell nuclei. A z-stack of 14 confocal microscopic images were taken and combined to a maximum intensity projection. Protein E fluorescence was quantified and normalized to the absence of SP in three z-stacks ± standard deviation (see Supplementary Figs. 8c, d and 9c, d). *P < 0.01, **P < 0.001, ***P < 0.0001 (by one-way ANOVA with Bonferroni post-test)

preparation only marginally lost antiviral activity when boiled (Fig. 5d). This is in agreement with only a small shift in the size distribution (Supplementary Fig. 11c) of EVs and only marginal effects on vesicle morphology (Supplementary Fig. 11d) upon boiling. This result suggests that the functionality of this EV preparation might not depend on protein or RNA constituents.

Confirming an effect on virion particle binding to target cells, EV preparation inhibited ZIKV attachment with similar efficiency as SP (Fig. 5e, f and Supplementary Fig. 12). In comparison, EV preparations derived from urine and saliva that are of similar or slightly larger sizes did not affect ZIKV infection (Supplementary Fig. 13). In conclusion, the SP fraction containing abundant EVs

**Table 2 Anti-ZIKV activity of individual ($n = 10$) SE samples and pooled SE**

| | | IC$_{50}$ [% ±SD] average | IC$_{50}$ [%] pooled SE | IC$_{50}$ [%] Min. | IC$_{50}$ [%] Max. |
|---|---|---|---|---|---|
| ZIKV GWUH | Cell treat. | 0.74 ± 0.18 | 0.67 | 0.49 | 0.98 |
| | Virion treat. | 1.02 ± 0.33 | 0.71 | 0.65 | 1.84 |
| ZIKV MR766 | Cell treat. | 0.84 ± 0.11 | 0.76 | 0.71 | 1.11 |
| | Virion treat. | 1.00 ± 0.25 | 0.92 | 0.68 | 1.39 |

Shown are IC$_{50}$ values derived from experiments shown in Fig. 4a, b and Supplementary Fig. 7a, b. Average IC$_{50}$ values represent means derived from the 10 donors; Min., lowest IC$_{50}$ measured; Max., highest IC$_{50}$ measured

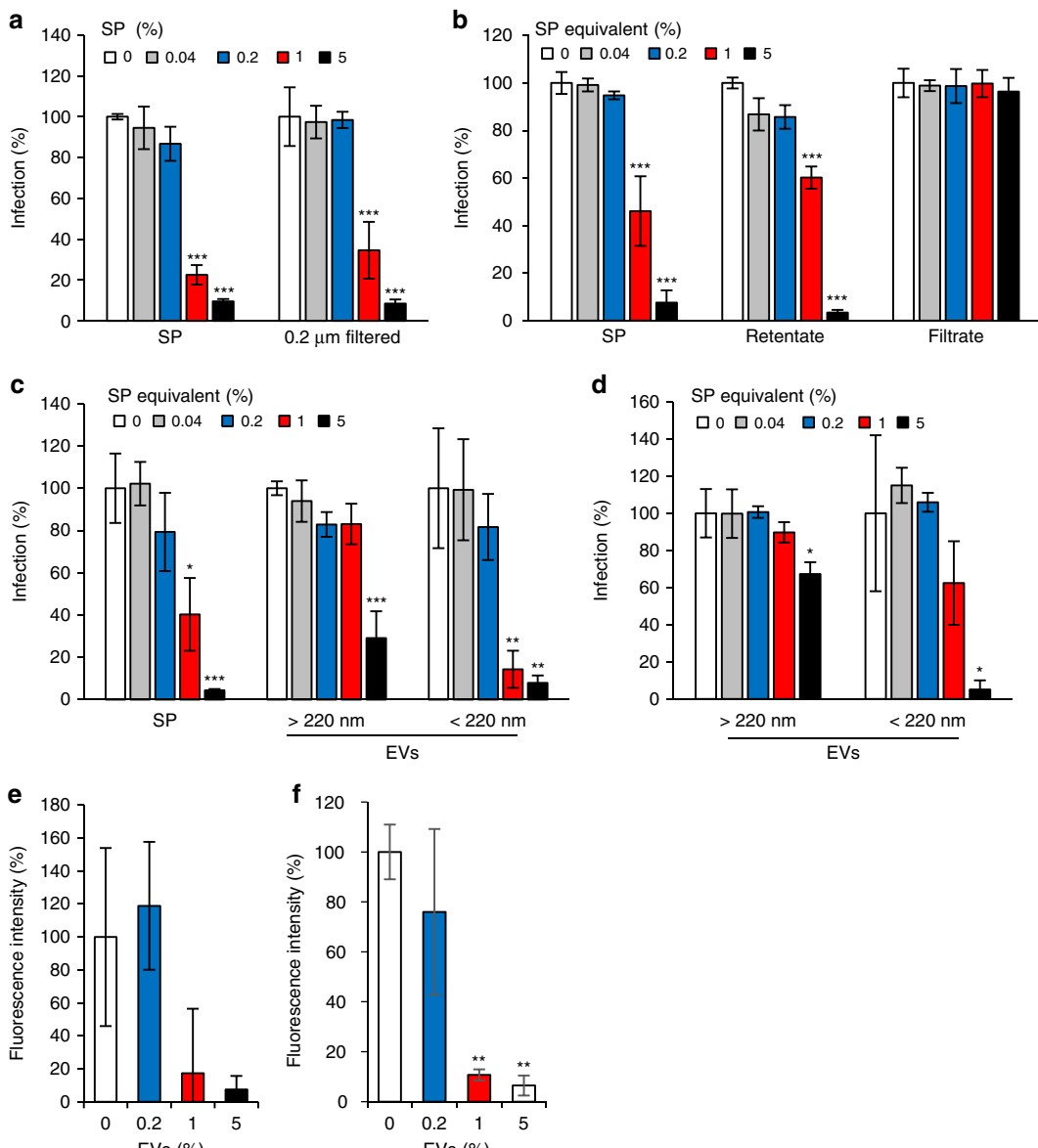

**Fig. 5** Seminal extracellular vesicles inhibit ZIKV infection. **a** 50% SP was filtered through a 0.2 μm syringe filter. **b** 0.2 μm filtered SP was applied to a 300 kDa molecular weight filter and the retentate and filtrate diluted with PBS to the originally applied volume. **c** SP was separated into extracellular vesicles (EVs) larger and smaller than 220 nm by centrifugal size filtration. **d** EV samples generated in **c** were boiled at 99 °C for 20 min, centrifuged, and denatured protein discarded. All samples (**a**–**d**) were added to Vero E6 cells at the indicated concentrations and incubated for 10 min before cells were inoculated with ZIKV MR766. After 2 days, infection was determined by cell-based ZIKV immunodetection assay that enzymatically quantifies the flavivirus protein E. Average infection rates are normalized to the corresponding infection averages in the absence of SP. Data represent average values obtained from triplicate infections ± standard deviations. **e** ZIKV virions were allowed to attach to Vero E6 cells in the presence of the indicated concentrations of SP-derived EVs (<220 nm) for 2 h at 37 °C or **f** 4 °C. Cells were then washed and stained for ZIKV protein E and cell nuclei. A z-stack of 14 confocal microscopic images were taken and combined to a maximum intensity projection. Protein E fluorescence was quantified and normalized to the absence of EVs in three z-stacks ± standard deviation (see Supplementary Fig. 12). *$P < 0.01$, **$P < 0.001$, ***$P < 0.0001$ (by one-way ANOVA with Bonferroni post-test)

is responsible for the observed anti-ZIKV activity of SE. However, the exact nature of the antivirally active constituent in the EV preparations from SE remains to be determined.

**A fresh ejaculate inhibits ZIKV**. The results presented thus far were generated using freeze/thawed SE or SP. To exclude that the freeze/thaw process is responsible for the observed anti-ZIKV activity, we tested a freshly derived liquefied ejaculate (SE) and SP derived from this fresh sample. Fresh SE and SP both efficiently blocked ZIKV MR766 and GWUH infection of Vero E6 cells under both "cell treatment" (Supplementary Fig. 14a) and "virion treatment" (Supplementary Fig. 14b) conditions. Again, 1% SE or SP reduced ZIKV infection by >90%, and infection was almost entirely prevented by 5% SE or SP, corroborating results obtained with frozen samples.

**SE inhibits Dengue and West Nile virus infection**. We next tested whether SE may also inhibit infection by Dengue virus (DENV), another mosquito-transmitted flavivirus. For this, Vero E6 cells were infected with ZIKV or DENV in the presence of SE or SP, and after 2 days, intracellular E protein expression levels were measured by the in-cell immunodetection assay. SE and SP both inhibited ZIKV (Fig. 6a) and DENV (Fig. 6b) infections with similar efficiencies. These results were confirmed in Huh-7 cells using a DENV reporter virus expressing Renilla luciferase (Supplementary Fig. 15a, b). We further demonstrated that, similar to

ZIKV (Fig. 2), DENV infection was not affected by SEVI fibrils (Supplementary Fig. 15c, d). Finally, we demonstrated that SE also effectively inhibited infection by West Nile virus (WNV) (Fig. 6c). Thus, SE suppressed infection by ZIKV, DENV, and WNV.

**SE enhances HIV-1 while inhibiting ZIKV infection**. In contrast to ZIKV, DENV, and WNV, HIV-1 infection is enhanced by SE or SP[38–40]. To make sure differences in the performance or variation in SE batches did not account for the seemingly opposite effects of SE on HIV-1 and flavivirus infection, we next tested the effect of SE on ZIKV and HIV-1 simultaneously under the same experimental set-up and conditions. An MOI of ~0.05 of ZIKV or HIV-1 was incubated with 50% SE or serial five-fold dilutions thereof. The SE-treated virions were added to TZM-bl cells, which are permissive for both HIV-1 and ZIKV (Supplementary Fig. 2b). After 2 h, the inoculum was removed and fresh medium was added. HIV-1 infection rates were determined quantifying cell-associated β-galactosidase[57,61] and ZIKV infection by the in-cell immunodetection assay. As expected[38], SE markedly increased HIV-1 infection with a maximum enhancement of 23-fold when virions were pre-exposed to 10% SE (Fig. 6d, e). In contrast, ZIKV infection was efficiently suppressed when the same experimental set-up was applied and almost entirely blocked by 50% SE (Fig. 6d, e). Thus, the identical SE batch and treatment conditions enhance infection by HIV-1 while inhibiting infection by ZIKV.

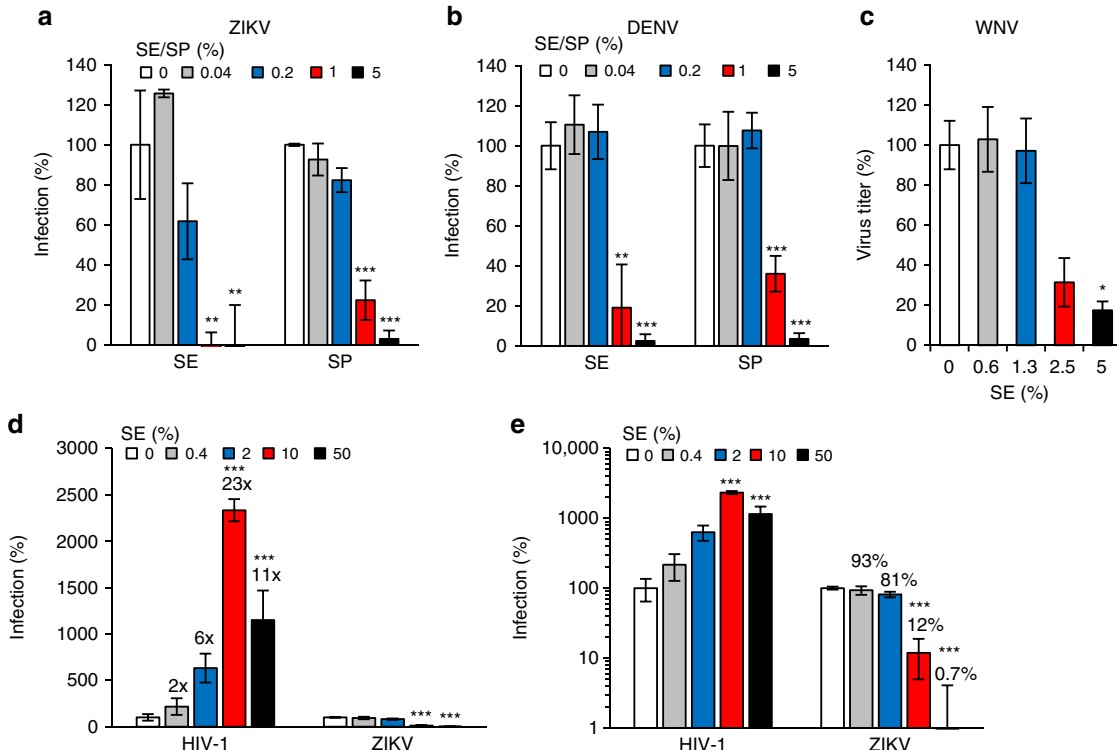

**Fig. 6** Semen and seminal plasma inhibit Dengue and West Nile virus but enhance HIV-1 infection. Vero E6 cells were incubated with the indicated concentrations of SE or SP and inoculated with **a** ZIKV MR766 or **b** DENV-2 (Thailand/16681/84). Two days later, infection rates were determined by a cell-based ZIKV and DENV immunodetection assay that enzymatically quantifies the flavivirus protein E. **c** WNV NY99 was used to infect Huh-7 cells in the presence of the indicated concentrations of SE. Twenty four hours later, supernatants were collected and used to detect viral titers in a plaque assay. **d** Similar MOIs of ZIKV GWUH and HIV-1 NL4-3 92TH014.12 were incubated with 0, 0.4, 2, 10, or 50% of SE for 10 min. Vero E6 or TZM-bl cells were then inoculated with the SE-treated virions. Two days later, infection rates were determined by a cell-based ZIKV immunodetection assay or by quantifying β-galactosidase activity of cell lysates for HIV-1. Data are plotted on a linear y axis. **e** Data from **d** are plotted on a logarithmic y axis to better visualize inhibition of ZIKV infection. Numbers above the columns indicate fold enhancement of infection levels. Data are normalized to infection rates in the absence of SE or SP and represent average values obtained from triplicate (WNV: duplicate) infections ± standard deviations. *$P < 0.01$, **$P < 0.001$, ***$P < 0.0001$ (by one-way ANOVA with Bonferroni post-test)

**SE inhibits infection of anogenital cells and tissues**. We then tested whether SP is capable of inhibiting ZIKV infection of cells of the anogenital tract. We used SP instead of SE so that endogenous cells would not hamper the visualization of infected cells by confocal microscopy. We observed that SP concentrations of 1% markedly reduced infection of primary HFF and the HeLa, SW480, and SKOV3 cell lines (Fig. 7a, b). A final SP concentration of 5% completely inhibited ZIKV infection of primary eSFs, and a SP concentration of 25% completely inhibited infection in all analyzed cells (Fig. 7a, b). Cellular morphology as well

as staining of the cytoskeleton and nuclei were similar for uninfected and SP-exposed cells, excluding the possibility that cytotoxic effects of SP contributed to reduced ZIKV infection rates.

Finally, we determined whether SE can block ZIKV replication in vaginal explants. Explants from two donors were cut into 48 or 96 tissue blocks, respectively, divided into two equal parts in 15 ml tubes, and exposed to ZIKV in the absence or presence of SE (25%). After 90 min, blocks were thoroughly washed and each block was cultured individually in 96-well plates. Viral loads were determined immediately (wash control) and after 4 (tissue 1) or

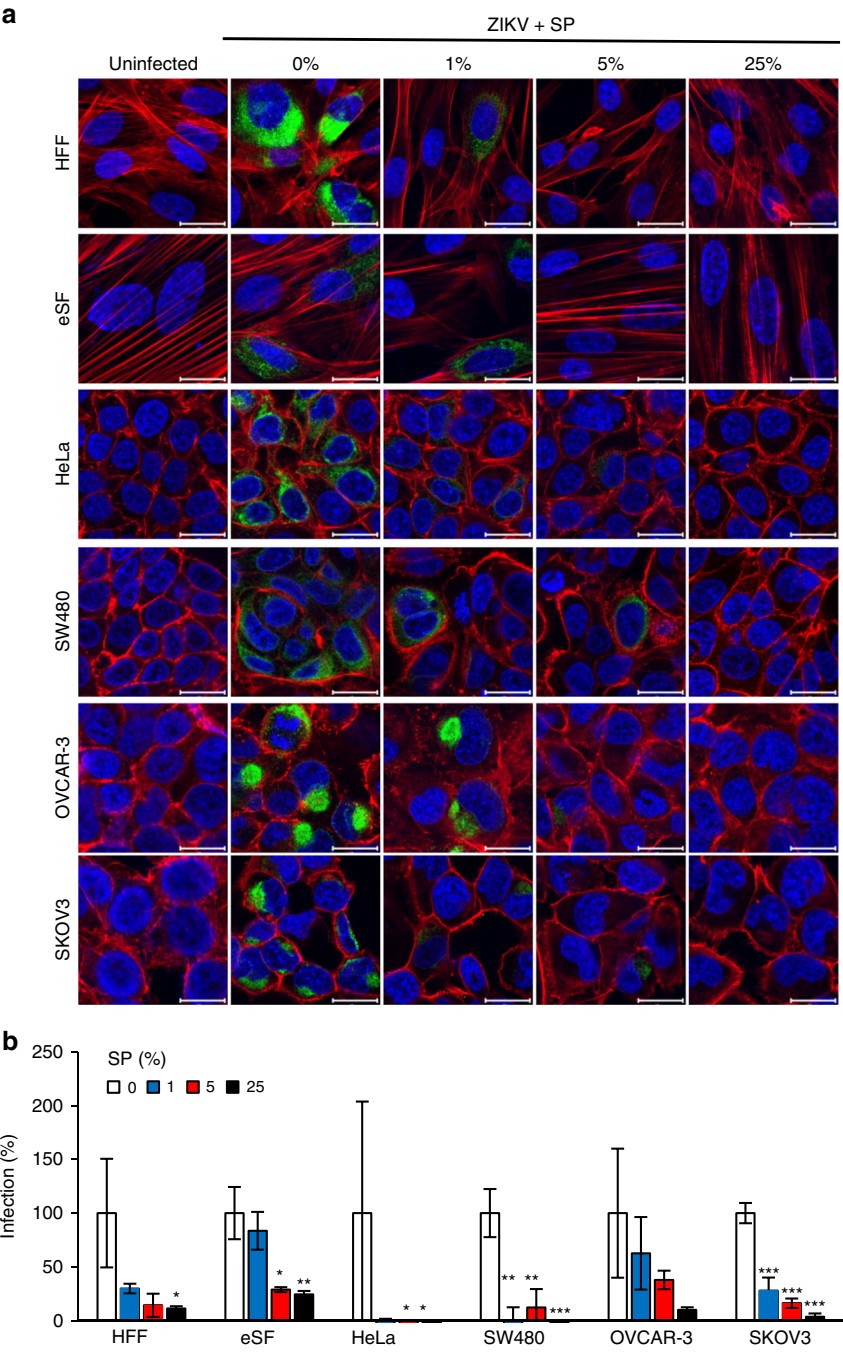

**Fig. 7** Seminal plasma inhibits ZIKV infection of cells of the anogenital tract. **a** Primary HFFs and eSFs and cell lines derived from the cervix (HeLa), colon (SW480), or ovaries (OVCAR-3 and SKOV3) were incubated with the indicated concentrations of SP and inoculated with ZIKV MR766. Two days later, cells were stained for protein E (green), nuclear DNA (blue), and actin (red) and then imaged by confocal microscopy. Scale bars correspond to 20 μm. **b** Fluorescence signal intensity was quantified and normalized to infection in absence of SP from three images ± standard deviation. HFF: human foreskin fibroblast, eSF: endometrial stromal fibroblast, *P < 0.01, **P < 0.001, ***P < 0.0001 (by one-way ANOVA with Bonferroni post-test)

5 days (tissue 2). Consistent with Supplementary Fig. 1, ZIKV replicated in the majority of the individual tissue blocks (Fig. 8a, b). Importantly, in the presence of 25% SE, viral loads were reduced on average by 79% ($P = 0.0198$, two-way repeated-measures (RM) analysis of variance (ANOVA)) (Fig. 8a) and 73.4% ($P = 0.0326$, two-way RM ANOVA) (Fig. 8b), respectively. Compared to infection experiments in cell lines and primary cells where 25% SE suppressed ZIKV entirely (Fig. 7), the antiviral efficacy of SE in tissues was slightly reduced, which is likely due to the higher virus inoculum required to achieve reliable infection and viral spread over the course of 4–5 days. A cytotoxicity assay performed with tissue blocks of donor A at day 8 revealed no signs of reduced viability in the presence of SE (Fig. 8c). Thus, SE suppresses productive ZIKV replication in primary VTs.

## Discussion

In this study, we used in vitro and ex vivo systems to characterize sexual transmission of ZIKV. It has been calculated that, upon male-to-female sexual contact, up to ~$10^8$ infectious ZIKV particles from a viremic individual (present in an average ejaculate volume of ~3.7 ml) may be deposited into the FRT[65]. Considering the high viral titers in SE and our demonstration that cell lines, primary cells, and tissues derived from the anogenital region efficiently support productive ZIKV infection, it is surprising that the rates of sexual ZIKV transmission are low compared to mosquito-mediated transmission, where typically only $10^2$ infectious virions are injected intravascularly[65,66]. Our results reported here suggest that the reason why ZIKV has not turned into a sexually transmitted epidemic may be explained by the fact that SE is a powerful inhibitor of ZIKV infection. SE prevented ZIKV infection of cells derived from the anogenital tract including VT, suggesting that this intrinsic anti-ZIKV activity of SE may also prevent male-to-female genital ZIKV transmission in vivo.

ZIKV has previously been reported to infect skin fibroblasts, keratinocytes, dendritic cells[55], placental cells, and neuronal progenitors[67]. We here confirm that ZIKV also replicates in isolated eSFs[68] and show that the tropism extends to cervix-, colon- and foreskin-derived cells (Fig. 1) and most notably to explant tissues derived from endometrium and vagina (Fig. 2 and Supplementary Fig. 1). These explants largely resemble the cyto-architecture present in vivo and allow for the ex vivo study of viral replication in three-dimensional cultures. Our findings further underline the notion that ZIKV has a broad cellular tropism

and that permissive cells are located at potential entry sites of sexual transmission.

Inhibition of ZIKV infection was observed with pooled SE derived from multiple donors (Figs. 3, 6, 8 and Supplementary Figs. 4, 6), with SE from individuals (Fig. 4 and Supplementary Fig. 7), and with a fresh ejaculate (Supplementary Fig. 14), suggesting that viral inhibition is a general property of this bodily fluid. The anti-ZIKV effect is mediated by a soluble factor because SP, the cell-free supernatant of SE, also markedly suppressed infection (Figs. 3, 4, 5, 6, 7, and Supplementary Figs. 5, 6, 10). The factor reversibly inhibits an early step in the viral life cycle, since adding SE after infection has been initiated does not limit viral replication (Fig. 4c). This antiviral activity is directed toward the cell since the final cell culture concentration (and not concentrations during virion treatment) determined the magnitude of the anti-ZIKV effect of SE (Fig. 4, and Supplementary Figs. 7, 14). Consistent with these observations is our finding that SP abrogates the attachment of virions to target cells (Fig. 4d, e and Supplementary Fig. 8c, d and 9c, d).

The exact nature of the constituent in SE responsible for ZIKV inhibition remains to be determined. Our results show that this factor is present in EV preparations (Fig. 5c, d). We found that a SE fraction containing EVs ~<220 nm in diameter inhibit ZIKV infection as efficiently as SP (Fig. 5c), whereas EV preparations from urine and saliva had no effect (Supplementary Fig. 13). Moreover, the SP-derived preparation also prevented ZIKV attachment (Fig. 5e, f and Supplementary Fig. 12), as observed for SP (Fig. 4d, e and Supplementary Figs. 8c, d and 9c, d), and contained EVs with an average diameter of ~142 nm (Supplementary Fig. 11a) that are largely thermoresistant (Supplementary Fig. 11b, c). In contrast, a SP fraction devoid of EVs did not abrogate ZIKV infection (Fig. 5b and Supplementary Fig. 11b). EVs are small vesicles released by cells and are common components of many bodily fluids[69]. Human SE contains an extra-ordinary high concentration of EVs with an average number of ~$1 \times 10^{13}$ vesicles per ml SE[70]. A recent cryo-electron microscopic analysis of human SE revealed 11 subcategories of EVs, including multifaceted assemblies, with diameters between ~20 and 400 nm[69]. The great morphological diversity accompanies the functional diversity of seminal EVs, which are involved in sperm motility, SE liquefaction, immunosuppression in the female genital tract to avoid anti-spermatozoa immunity, and prevention of microbial infections[69]. Which particular type or constituent in the

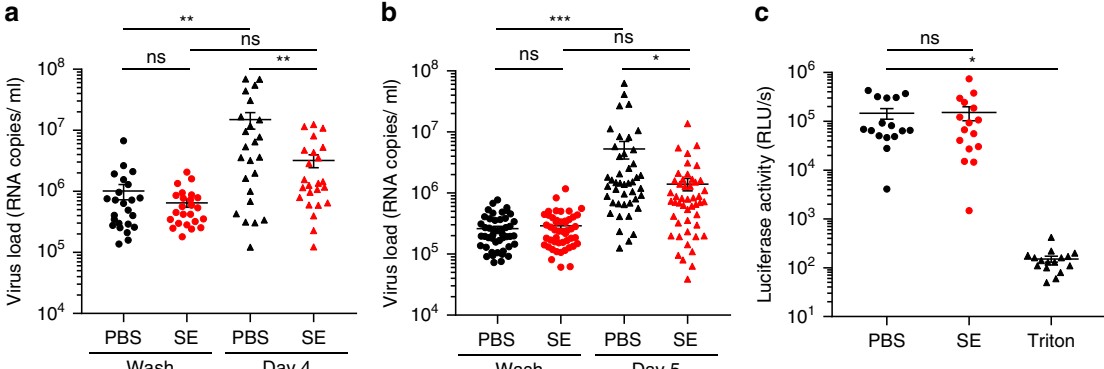

**Fig. 8** Semen reduces ZIKV infection of primary vaginal tissue. **a** 2 × 24 vaginal tissue blocks of donor A, and **b** 2 × 48 blocks of donor B were incubated for 90 min with ZIKV GWUH in the presence of 25% semen or 25% PBS. After washing, individual blocks were transferred into separate wells of a 96-well plate and incubated at 37 °C. Aliquots were taken at day 0 (wash control) or day 4 or 5 and examined for viral genome copy numbers by qPCR. *$P < 0.05$, **$P < 0.01$, ***$P < 0.001$ (by two-way RM ANOVA with Tukey's multiple comparison post-test). **c** 3 × 16 vaginal blocks were treated with PBS, 25% semen, or 0.05% triton for 90 min. After washing, blocks were incubated in separate 96 wells for 8 days and viability was examined by CellTiter-Glo® luminescent cell viability assay. Black lines indicate averages ± standard error of the mean, RLU/s relative light units per second. ns: not significant, *$P < 0.05$ (by one-way ANOVA with Bonferroni post-test)

EV preparations exerts anti-ZIKV activity and the precise mechanisms employed needs to be clarified in subsequent studies. One plausible explanation is, however, that the cellular target of the functional entity is a ubiquitously expressed receptor or attachment factor, which is utilized by ZIKV for infection. Moreover, this cellular structure is conserved between African green monkeys and humans, as SE blocks infection of cells derived from both species (Figs. 3, 7). The identification of this conserved cellular target will not only explain how SE might prevent ZIKV infection but may also identify a novel attachment factor, which is necessary for not only ZIKV but also DENV and WNV entry.

Our finding that SE and SP inhibit ZIKV infection came as a surprise. We and others have previously shown that SE enhances HIV-1[38], HSV-2[58], and CMV[59] infection. This infectivity-promoting effect was attributed to positively charged amyloid fibrils in SE which concentrate the virions that are encased by a negatively charged membrane. This in turn increases viral attachment to and fusion with cellular targets[43]. However, synthetic SEVI fibrils neither enhanced nor inhibited ZIKV infection, suggesting that seminal amyloid does not interact with the ZIKV particle. This could be explained by the fact that the Zika virion is covered by a dense coat of the viral E protein, which renders the viral lipid membrane largely inaccessible to large external factors[71], such as seminal fibrils. In contrast, the membrane of HIV-1 is largely accessible, because only a few viral glycoproteins are embedded[72], allowing efficient interaction of fibrils with HIV-1 particles. Further studies with viruses containing well-defined numbers of viral glycoproteins are needed to clarify whether the accessibility of the viral membrane determines fibril-mediated enhancement of viral infection. In addition, the effect of seminal amyloids on other emerging and re-emerging viruses should be determined to assess its potential to be transmitted via sexual intercourse.

With the realization that SE is not simply a passive carrier of ZIKV, but rather significantly inhibits infection rates in vitro, animal studies are urgently needed to determine the antiviral efficacy of SE on vaginal or rectal ZIKV infection, in order to better predict whether SE is likely to play a role in preventing sexual ZIKV transmission in humans. Another intriguing question is whether ZIKV is capable of acquiring resistance against the inhibitory factor(s) in SE that may increase the frequency of sexual transmission to a level allowing spread of this human pathogen in areas where no mosquitoes are present. Finally, it would be of interest to evaluate whether SE may exert similar effects on other flavi- or arthropod-borne viruses. Our results presented here show that SE also restricts DENV and WNV infection. Sexual transmission of hepatitis C virus is an extremely infrequent event[73] and has never been recorded for Dengue, West Nile, yellow fever, or Chikungunya virus, although these viruses may be shed into SE of infected individuals[74–76]. Clarifying whether SE has a broad-based anti-flavivirus activity may help explain why certain pathogens are typically not, or only very infrequently, transmitted via sexual intercourse.

## Methods

**Cell and tissue culture**. Vero E6 (*Cercopithecus aethiops*-derived epithelial kidney) (ATCC, CRL-1586) cells were grown in Dulbecco's modified Eagle's medium (DMEM) supplemented with 2.5% inactivated fetal calf serum (FCS), 2 mM L-glutamine, 100 units/ml penicillin, 100 μg/ml streptomycin, 1 mM sodium pyruvate, and non-essential amino acids (Sigma #M7145). HeLa (human epithelial cervix carcinoma) (ATCC, CCL-2), TZM-bl (HeLa-based HIV-1 reporter) (ARRRP, 8129), and SKOV3 (human epithelial ovary carcinoma) cells (ATCC, HTB-77) were grown in DMEM supplemented with 10% FCS, 2 mM L-glutamine, 100 units/ml penicillin, and 100 μg/ml streptomycin. Huh-7 (differentiated hepatocyte-derived carcinoma) cells (authenticated by SNP profiling at Multiplexion) were grown in DMEM supplemented with 10% FCS, 100 units/ml penicillin, 100 μg/ml streptomycin, and non-essential amino acids. SW480 (human epithelial colon carcinoma) (ATCC, CCL-228) and OVCAR-3 (human epithelial

ovary carcinoma) cells (ATCC, HTB-161) were grown in RPMI-1640 supplemented with 10% FCS, 2 mM L-glutamine, 100 units/ml penicillin, and 100 μg/ml streptomycin. HFFs were isolated from tissue samples that were residuals from routine procedures and obtained anonymized after written informed consent of the parents in agreement with articles 21 and 23 of the recommendations of the council of Europe (2006). HFFs were grown in minimal essential medium (MEM) supplemented with, 2 mM L-glutamine, 10% FCS and non-essential amino acids, 100 units/ml penicillin, and 100 μg/ml streptomycin. T84 (human epithelial colon carcinoma) cells (ATCC, CRL-248) were grown in a 1:1 mixture of Ham's F12 medium and DMEM supplemented with 2 mM L-glutamine, 5% FCS, 100 units/ml penicillin, and 100 μg/ml streptomycin. Primary human eSFs were obtained from the Cooperative Human Tissue Network (CHTN) (IRB # 14-15361) and isolated and cultured as described (http://www.bio-protocol.org/e1623)[77] in SCM medium consisting of 67.5% DMEM with 1 mM sodium pyruvate, 22.5% MCDB-105, 5 μg/ml insulin, 10% FCS, 100 units/ml penicillin, and 100 μg/ml streptomycin. C6/36 (*Aedes albopictus*) mosquito cells (ECACC, 89051705) were grown in Leibovitz L-15 medium (Gibco) supplemented with 10% FCS, 100 units/ml penicillin, 100 μg/ml streptomycin, and 10 mM 4-(2-hydroxyethyl)-1-piperazineethanesulfonic acid. Blocks cut from surgically removed tissue of cervix and vagina of pelvic organ prolapse patients who gave informed consent (approved by the ethics committee of Ulm University) were cultured in extracellular medium (ECM) consisting of RPMI with 15% FCS, 2 mM L-glutamine, 1 mM sodium pyruvate, non-essential amino acids, 100 units/ml penicillin, 100 μg/ml streptomycin, 100 μg/ml gentamicin, and 25 μg/ml Amphotericin B. All experiments in the presence of SE and SP were performed in the presence of 100 μg/ml gentamicin to prevent bacterial outgrowth. Mosquito cells were cultured at 28 °C and human and monkey cells and tissues were cultured at 37 °C in a 5% $CO_2$ humidified incubator. All cell lines were regularly tested for mycoplasma contamination.

**SE and SP**. SE from healthy donors was obtained from the "Kinderwunsch-Zentrum Ulm", a fertility center in Ulm, or healthy volunteers at Ulm University, after informed consent had been given (approved by the ethics committee of Ulm University). Ejaculates were allowed to liquefy for 30 min and then stored at −80 °C as individual or pooled samples. SP was generated by centrifugation of SE for 30 min at 20,000 × $g$ at 4 °C and collection of the cell-free supernatant.

**Viruses**. Plasmid pBRNL4-3 92TH014.12 encodes the HIV-1 NL4 3 provirus in which the V3-loop region is replaced by the V3-loop of the CCR5-tropic 92th014.12 isolate[39,41,43,61]. ZIKV strain MR766 was isolated in 1947 from a sentinel rhesus macaque[1]. FB-GWUH-2016 is a ZIKV strain that was isolated in 2016 from a fetal brain with severe abnormalities[54]. DENV-2 (Thailand/16681/84) cDNA was constructed from a serum-derived virus[78]. This virus was used to construct the renilla luciferase encoding DENV-R2a[79]. WNV NY99 (385-99) was isolated from the liver of a snowy owl (*Nyctea scadiaca*)[80].

**Virus stock generation and propagation**. ZIKV was propagated by inoculation of 70% confluent Vero E6 cells in 175 cm² cell culture flasks for 2 h in 5 ml medium. Subsequently, 35 ml fresh medium was added and the cells were cultured for 3–5 days. CPE was monitored by light microscopy and virus was harvested when 70% of the cells detached owing to CPE. Supernatants were centrifuged for 3 min at 325 × $g$ to remove cellular debris and then aliquoted and stored at −80 °C as virus stocks. $TCID_{50}$ of each stock was determined by infection of Vero E6 cells with serially diluted virus stocks and calculated according to Reed and Muench. The genome copy number of the stocks was assessed by quantitative reverse transcriptase-polymerase chain reaction (RT-qPCR; RealStar® Zika Virus RT-PCR Kit, Altona Diagnostics, Hamburg, Germany) (see Supplementary Table 1).

DENV-R2a was propagated in Vero E6 cells. DENV-2 (Thailand/16681/84) and WNV NY99 (and additionally for one experiment ZIKV MR766) were prepared by virus amplification in insect C6/36 cells.

HIV-1 stocks were generated by transient transfection of HEK293T cells using the calcium phosphate precipitation technique. HEK293T cells were seeded in 6-well plates or 175 cm² flasks, and the following day, when cells were at a confluence of 60–80%, cells were replaced with fresh medium. A total of 5 μg (for 2 ml medium per 6 well) or 125 μg (for 50 ml medium per 175 cm² flask) of proviral plasmid DNA was diluted in 125 mM $CaCl_2$ and added dropwise to an equal volume of 2× HBS. After 10 s of vortexing, the DNA precipitate was added dropwise to the culture medium. Sixteen hours post-transfection, medium was replaced with fresh DMEM (supplemented with 2.5% FCS) and incubated for another 24 h. The supernatant, containing infectious virions, was centrifuged (3 min, 325 × $g$) to remove cellular debris and transferred into reaction tubes. Virus stocks were stored at 4 °C or frozen at −80 °C.

**$TCID_{50}$ end point titration of ZIKV**. To determine infectious ZIKV titer, 6000 Vero E6 cells were seeded per well in 96 flat-bottom well plates in 100 μl medium and incubated overnight. The next day, 80 μl fresh medium was added. For end point $TCID_{50}$ determination, ZIKV samples were titrated 10-fold, and 20 μl of each dilution was used for inoculation of Vero E6 cells. This end point titration resulted in final ZIKV dilutions of $10-10^9×$ on the cells in triplicates or sextuplicates. Cells

were then incubated and monitored by microscopy for CPE and plaque formation. TCID$_{50}$/ml was calculated according to Reed and Muench.

**CellTiter-Glo® luminescent cell viability assay**. CellTiter-Glo® luminescent cell viability assay (Promega #G7571) was performed according to the manufacturer's instructions. Briefly, medium was removed from the cells or tissue blocks, and 50 μl phosphate-buffered saline (PBS) and 50 μl of reagent were added. After a 10 min incubation, luminescence was measured in an Orion II Microplate Luminometer (Titertek Berthold). Untreated controls were set to 100% viability. Error bars are standard deviations of triplicates.

**Fluorescence microscopy of ZIKV infection**. Target cells were seeded in eight-well μ-Slides (Ibidi) and incubated overnight. Cells were inoculated with $10^3$–$10^4$ TCID$_{50}$/ml of ZIKV. After 2 h, the inoculum was removed and cells washed with PBS before fresh medium was added. After 2 days of culture, cells were washed with PBS, fixed with 4% paraformaldehyde in PBS for 10 min at 4 °C, permeabilized with 0.1% (v/v) Triton X-100 in PBS for 5 min, and washed again. Unspecific binding sites were blocked by 30 min incubation with 5% (v/v) FCS and 1% (v/v) bovine serum albumin (BSA) in PBS. Cells were stained with 1:10,000 diluted mouse anti-flavivirus group antigen/protein E antibody (4G2) (absolute antibody, Ab00230-2.0) in PBS with 1% (v/v) BSA for 45 min. After 3 washes with PBS, cells were incubated with 1:1000 diluted goat anti-mouse secondary antibody conjugated to Alexa Fluor 488 (ThermoFisher Scientific, A11001) for 45 min and stained with 0.3 μg/ml 4′,6-diamidino-2-phenylindole dihydrochloride (Sigma, D9542). Alternatively, the nuclei were stained with 1:2000 diluted Hoechst 33342 (ThermoFisher Scientific, H1399) together with staining of actin filaments by 1:400 diluted Phalloidin-Atto 647 N (ATTO-TEC, AD 647N-82). Slides were imaged by using an inverted fluorescence microscope. Images were generated from $3 \times 3$ individual images taken with a 20× objective lens of an Axio-Observer.Z1 fluorescence microscope, the Axiovision 4.8 software and the Mosaic software module (Zeiss). Infected cells were counted using ImageJ with the ITCN (Image-based Tool for Counting Nuclei) plugin. Phalloidin-stained samples were imaged by confocal microscopy using a Zeiss LSM 710 and images processed using the ZEN software 2010, and infection intensity was quantified by measuring the total signal intensity per image and subtracting the uninfected background using the ZEN software 2010.

**ZIKV infection of VT and ET blocks**. VT and ET preparation and infection was performed similar to HIV-1 cervical tissue explant studies[52,53]. Tissues were cut into $2 \times 2 \times 1$ mm$^3$ tissue blocks. Six blocks were inoculated with ZIKV MR766 or GWUH by incubation of the blocks individually in a 96-well containing 200 μl medium and virus stock at 37 °C. After 2 h, blocks were washed 3 times with PBS, and the 6 blocks together were transferred onto geofoams that were prepared in a 12-well plate containing 800 μl ECM culture medium. Alternatively, to detect viral replication in each single vaginal block, blocks were incubated with ZIKV in a 15 ml falcon, washed with PBS, and transferred into separate 96 wells containing 200 μl ECM culture medium. In all, 100 μl supernatant was collected immediately (wash control, day 0) or at consecutive days, stored at −80 °C, and analyzed by TCID$_{50}$ titration for infectious virus titer or by RT-qPCR for viral RNA copy numbers (see Supplementary Table 1).

**Cell-based virus immunodetection assay**. For infection, 6000 target cells were seeded per 96-well flat-bottom well in 100 μl appropriate medium and cultured overnight. After adding 80 μl of medium, cells were inoculated with $10^3$–$10^5$ TCID$_{50}$/ml ZIKV in triplicates. In experiments with SE or SP, medium was supplemented with 100 μg/ml gentamicin to prevent bacterial outgrowth and changed 2 h after infection to minimize cytotoxic SE effects[39,60,61]. For cell treatment, cells were incubated with 0, 0.04, 0.2, 1, and 5% SE and SP before inoculation with ZIKV. For virion treatment, virus particles were preincubated with ten-fold higher concentrations (0, 0.4, 2, 10, or 50%) of SE and SP before the mix was inoculated (and diluted ten-fold) on cells. Two days post inoculation, infection rates were determined by immunodetection of ZIKV-infected cells as already described[56]. Cells were rinsed with PBS, fixed for 20 min with 4% paraformaldehyde, permeabilized with cold methanol for 5 min at 4 °C, and washed with PBS. Next, they were incubated with 1:10,000 mouse anti-flavivirus group antigen/protein E antibody (4G2) (absolute antibody, Ab00230-2.0) or 1:1000 diluted anti-HIV-1 p24 antigen (mAK183) (EXBIO Antibodies) in PBS containing 10% (v/v) FCS and 0.3% (v/v) Tween 20 for 1 h at 37 °C. Following 3 washes, cells were incubated with a 1:20,000 diluted horseradish peroxidase-coupled anti-mouse antibody (Thermo-Fisher Scientific, A16066) for 1 h at 37 °C. Cells were then washed four times and tetramethylbenzidine peroxidase substrate was added. After incubation at room temperature for 5 min, the reaction was stopped with 0.5 M sulfuric acid. Absorption was measured at 450 nm and the baseline was corrected at 650 nm using a VMax Kinetic ELISA microplate reader. Values were corrected for the background signal derived from uninfected cells, and triplicates were expressed as average infection rates ± standard deviation.

**Flow cytometry of ZIKV-infected cells**. Six thousand Vero E6 cells were seeded per 96 well and cultured overnight. The next day, cells were inoculated with SE

samples and ZIKV. After 2 h, the inoculum was removed, cells were washed with PBS, and fresh medium was added. Two days later, cells were prepared for flow cytometric analysis to determine cell death and infection rates. Cells were detached from the well using 0.05% Trypsin/EDTA and resuspended in culture medium. Cells were then transferred to V-bottom 96-well plates, washed with PBS, and stained with fixable viability stain 450 for 15 min at room temperature. After a washing step, the cells were fixed and permeabilized with buffer A (fix & perm, Biozol, GAS002) for 15 min. Cells were then incubated with mouse anti-flavivirus group antigen/protein E antibody (4G2) (absolute antibody, Ab00230-2.0) diluted 1:10,000 in buffer B (fix & perm, Biozol GAS002) for 30 min at 4 °C. After a washing step, cells were incubated with 1:150 diluted goat anti-mouse secondary antibodies conjugated with Alexa Fluor 488 (ThermoFisher Scientific, A11001) in buffer B for 30 min protected from light. After further washing steps, cells were resuspended in fluorescence-activated cell sorting buffer and measured in a BD FACSCanto™ II Cell Analyzer. Unstained cells, single stained, and isotype controls served as a control.

**ZIKV virion attachment assay**. Target cells were seeded in eight-well μ-Slides (Ibidi, 80826) and incubated overnight. Cells were inoculated with ZIKV MR766 and incubated for 2 h at 37 °C or 4 °C. Subsequently, the inoculum was removed and cells washed with PBS. To detect virion attachment, cells were prepared immediately by immunofluorescence staining as described above. Attached virus particles were imaged as z-stacks of 14 images by confocal microscopy using a Zeiss LSM 710. Images were processed and combined to maximum intensity projections using the ZEN software 2010. Attached virions were quantified by measuring the total signal intensity per projection and subtracting the uninfected background using the ZEN software 2010.

**HIV-1 infection assay**. HIV-1 infection was detected by cell-based immunodetection assay (see above) or by the reporter activity of TZM-bl cells. This human cell line expressing CD4, CCR5, and CXCR4 encodes the *lacZ* gene under control of the HIV-1 LTR promoter[57]. Upon infection by HIV and simian immunodeficiency virus, the viral protein Tat induces β-galactosidase expression, allowing quantification of infection by conversion of a chemiluminescent substrate. To quantify HIV infection, 1 day prior to infection, 10,000 cells were seeded per well in 96-well flat-bottom microtiter plates in 100 μl medium. The next day, 80 μl medium was added and cells were inoculated with 20 μl sample in triplicates. Media was changed 2 h after inoculation to minimize cytotoxic SE effects (see http://www.bio-protocol.org/e1871)[61]. Three days postinfection, medium was discarded and 40 μl of diluted Gal-Screen® substrate/buffer A (1:8 in PBS) (ThermoFisher Scientific, T1027) was added. During the 30 min room temperature incubation, cells were lysed and the substrate converted by the released β-galactosidase. After incubation, 35 μl were transferred into white 96-well plates and substrate conversion was measured as relative light units per second using the Orion II Microplate Luminometer. Values were corrected for the background signal derived from uninfected cells, and triplicates were expressed as average infection rates ± standard deviation.

**Plaque assay of WNV**. VeroE6 were infected with serially diluted virus preparation in DMEM at 37 °C. Inoculum was removed 2 h after infection and cells were cultured in medium containing 1.5% carboxymethylcellulose. Cells were fixed 3 days postinfection by adding formaldehyde directly to the medium (5% final concentration). Cells were washed with water and stained with 1% crystal violet/10% ethanol for 30 min. Staining solution was removed and stained cells were rinsed again with water. Plaques were counted and titers of infectious virus were calculated.

**SP treatments**. Denaturation: SP was diluted 1:1 (v/v) in PBS and incubated at 99 °C for 20 min in an Eppendorf Thermomixer. Samples were then centrifuged for 15 min at $20,000 \times g$. The pellets were then discarded and the supernatants used for further analyses.

Proteinase K digestion: SP was diluted 1:1 (v/v) in PBS containing 200 μg/ml gentamicin and was incubated with 300 μg/ml Proteinase K (Roche, 03115887001) for 5 h at 37 °C. To stop proteinase K activity, the sample was denatured and centrifuged as described above.

Protein precipitation with 2,2,2,-tricholoroacetic acid (TCA): 0.2 μm syringe filtered SP (described below) was incubated with 10% TCA at room temperature for 1 h. Precipitated proteins were pelleted as described above and the supernatant adjusted to pH 7–8 with NaOH.

Lectin affinity chromatography: SP was applied to HiTrap™ Con A 4B (GE Healthcare, 28-9520-85) columns as described by the manufacturer. Briefly, the columns were washed and equilibrated with binding buffer at a flow rate of 0.2 and 1 ml/min. Next, SP was diluted 1:10 (v/v) in binding buffer and filtered through 0.2 μm syringe filters before it was applied to the columns at a flow rate of 0.1–0.5 ml/min. The filtrate was collected and the retentate eluted using 0.1 M and 0.5 M methyl-αD-glucopyranoside elution buffer. Buffers were exchanged by 3K Amicon centrifugal filters (see below) to PBS at the original volume of SP.

Size filtration: SP was diluted 1:1 (v/v) in PBS before it was applied to 0.2 μm syringe filters and the filtrate was used for further experiments.

Molecular weight filtration: SP was diluted 1:1 (v/v) in PBS and centrifuged at 14,000 × g for 30 min in Nanosep 300K Centrifugal Devices (Pall, OD300C33) or 3K Amicon Ultra-0.5 ml Centrifugal Filters (Millipore, UFC500324). The filtrate was collected and the retentate extracted by centrifugation at 1000 × g for 2 min. Volumes were adjusted with PBS to the original SP volumes.

EV isolation: EVs were isolated using Ultrafree-MC, DV 0.65 μm (Millipore, UFC30DV0S) and GV 0.22 μm (Millipore, UFC30GV0S) centrifugal filter units as described[64]. SP was centrifuged in the 0.65 μm filter unit for 20 min at 1000 × g, the filtrate was transferred to the 0.22 μm filter unit, and the procedure was repeated. The final filtrate contains the vesicles <220 nm. The two retentates were combined and diluted to the original SP volume and contain the vesicles >220 nm.

**Nanoparticle tracking analysis**. Videos of EVs derived from SP, urine, and saliva diluted in PBS were recorded for 60 s in a NanoSight LM10 and size distribution was evaluated by the NanoSight NTA software. Between the samples, the chamber was washed thoroughly and PBS was used to confirm absence of contaminating particles.

**Transmission electron microscopy**. Samples were adsorbed on glow discharged carbon-coated copper grids (Electric glow discharger Edwards High Vacuum) for 1 min at room temperature. Next, the grids were washed 3 times for 3 s in distilled H₂O and negatively stained with 2% (w/v) uranyl acetate in H₂O by another three incubation periods of 3 s. Excess solution was removed by filter paper and samples were allowed to dry. Samples were imaged with a JEOL JEM1400 transmission electron microscope at an accelerating voltage of 120 kV.

**Influence of SE on infection of VT blocks**. VT and ET preparation and infection was performed similar to HIV-1 cervical tissue explant studies[52,53]. Tissues were cut into 2 × 2 × 1 mm³ tissue blocks. Blocks were incubated with ZIKV GWUH and 25% SE or 25% PBS in a 15 ml falcon for 90 min. Blocks were then washed with PBS and transferred into separate 96 wells containing 200 μl ECM culture medium. In all, 100 μl supernatant was collected immediately (wash control, day 0) or at consecutive days, stored at −80 °C, and analyzed by qPCR for viral RNA copy numbers.

**Statistical analyses**. One-way ANOVA followed by Bonferroni's multiple comparison test (*$P < 0.01$, **$P < 0.001$, ***$P < 0.0001$), two-way RM ANOVA followed by Tukey's multiple comparison test (*$P < 0.05$, **$P < 0.01$, ***$P < 0.001$), and Friedman test was performed using GraphPad Prism version 7.03 for Windows, GraphPad Software, La Jolla, CA, USA, www.graphpad.com. Absence of significance (ns: not significant) was only indicated when relevant.

**Data availability**. All relevant data are available from the authors.

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

## Acknowledgements

We thank Martha Mayer for helping with tissue preparations and Stephanie Kallis for helping with the Dengue virus experiments. We thank the Kinderwunsch-Zentrum Ulm for providing semen samples and acknowledge the NIH Specialized Cooperative Centers Program in Reproduction and Infertility Research Human Endometrial Tissue and DNA Bank for endometrial tissues. Mirja Harms, Simone Joas, Rüdiger Groß, Manuel Hayn, Andrea Dietz, and Sina Lippold are part of the International Graduate School in Molecular Medicine Ulm. Anogenital cell lines were provided by Kerstin Otte (Institute of Applied Biotechnology, Biberach University of Applied Sciences, Biberach, Germany), Ninel Azoitei (Center for Internal Medicine I, University of Ulm, Ulm, Germany), or Markus Huber-Lang (Department of Orthopaedic Trauma, Hand-, Plastic- and Reconstructive Surgery, University Hospital of Ulm, Ulm Germany). J.A.M. is indebted to the Baden-Württemberg Stiftung for the financial support of this research project by the Eliteprogramme for Postdocs. N.R.R. acknowledges funding from the NIH (R21 AI122821 and R01 AI127219). J.M. acknowledges funding by the DFG (CRC1279).

## Author contributions

J.A.M. designed the experiments and Mi.H. established the cell-based immunodetection assay. Most experiments were performed by J.A.M. and Mi.H. F.K. performed confocal microscopy experiments. R.G. supported J.A.M. in seminal plasma treatments and EV experiments. S.J. performed staining and flow cytometry. A.D., S.L., and J.v.E. performed fluorescence microscopic experiments. Ma.H. supported J.A.M. and Mi.H. in immunodetection and TCID$_{50}$ assays. M.M. and A.S. performed qPCR analyses. M.C. performed WNV experiments. K.S.J. did EV preparations. Semen samples were provided by N.S.-M. Human tissues were obtained by F.E. and M.D. N.R.R. provided primary endometrial fibroblasts and seminal extracellular vesicle preparations and contributed to writing. M. O. provided NTA. B.M. advised statistics. R.B., J.-S.C., O.V., and J.-P.H. provided reagents, viruses, and contributed in writing the manuscript. J.M. designed the experiments, supervised work, and wrote the manuscript.
