## [Peer Review File · Nature Communications]

Reviewers' comments:

Reviewer #1 (Remarks to the Author):

The work of Müller et. al. is focused on the role of semen in Zika virus transmission. Sexual transmission of ZIKV is well established and therefore the importance of understanding mechanisms of this transmission and body defense systems are important. The authors used different experimental systems of Zika infection: cell lines primary cells from anogenital tract, explants of human cervix and proved that all of them support ZIKV replication. When they applied human semen, ZIKV replication in all these systems was inhibited. This was not due to the semen cytotoxicity because HIV replication in some of these systems was upregulated under semen treatment, confirming the authors' earlier observations.

The authors conclude that semen contains protective factors against ZIKV and present some data that these factors affect early stages of ZIKV infection.

Experiments are well-described, illustrated by convincing Figures and the data were subjected to appropriate statistical analysis.

I have the following critical remarks concerning the experiments and the way these experiments were interpreted:

1. Infection of ZIKV and HIV in TZM-bl cells were evaluated with different assays. While HIV infection was measured by fluorescence, the ZIKV infection was evaluated based on OD. How sensitive is the latter compared to the former. The authors claim that these cells were infectable by ZIKV. But could it be that the infection is so low and the assay is so insensitive that the stimulation can be missed? `

2. The authors claim that "Amongst the tested cell lines, viral replication was least efficient in SKOV3 and T-84 cells, and most efficient in SW480, HeLa, and OVCAR-3 cells". It is not clear how they compared the efficiency. Were the same number of cells infected? Was the cell density the same?

3. The authors report about significant variability of the explants in terms of ZIKV production. Did the authors normalize by the weight or other methods since it is extremely difficult to make the tissue blocks similar in size?

4. The description of the explants' culture should be made clearer. The authors should explain how "six blocks were incubated individually per 96 well" and how they are "transferred onto gelfoams in a 12 well plate".

5. Figure 3d. Did the authors normalize average by average or for every experiment?

6. P values should be added everywhere where the comparisons of means were performed.

Reviewer #2 (Remarks to the Author):

Dear Muller et al.

"Semen Inhibits Zika Virus Infection of Cells from the Anogenital Region"

In this manuscript, the authors showed that cells from anogenital region are susceptible to ZIKV. Since previous papers showed that ZIKV can be sexually transmitted from male to female, they examined whether semen affects the virus infectivity in these cells.

Interestingly, SEVI, which is known to boost HIV infection, did not change virus infectivity, but semen and seminal plasma were shown to inhibit ZIKV infection.

The inhibitory effect of semen against ZIKV is interesting, however, there are a few key questions that should be asked in order to decipher the responsible mechanisms (e.g. identifying key molecules).

Overall, the data are descriptive. Therefore, I think this manuscript is better suited for a more topic-specific journal.

Major points:

1. The authors should examine the mechanisms underlying why semen inhibits ZIKV infection. For example, they should try to identify which molecules (proteins, sugar, lipids, etc.) play an important role in the inhibition of ZIKV infection.
2. Fig.3c and d. They claim that SEVI did not enhance ZIKV infection in vitro. However, it seems that GWUH infectivity is enhanced after 50ug/ml SEVI treatment compared with 0 ug/ml SEVI treatment.
3. They showed that primary vaginal and endometrial tissue is susceptible to ZIKV (Fig. 2). So, testing the inhibitory effect of semen against ZIKV (related to Fig. 9) in these primary tissue cells would strengthen their conclusion.
4. Although few case of sexual transmission of flavivirus other than ZIKV have been reported, it would be informative to test whether semen also suppresses other flaviviruses such as dengue virus.

Specific points:

5. They should show more thorough statistical analysis on their data. Without proper statistical analysis, it is difficult to decipher the impact of some of their experiments.
6. Fig. 2. They only showed the highest detected virus titers, but they should show all data related to the replication of ZIKV in cells isolated from VT and ET.
7. Why are there no error bars in Fig. 4 c-e? Have they done only one experiment?
8. The experiment in Fig.5 is confusing. Usually CPE is not visible when virus cannot replicate/spread efficiently in the cells. It seems that this data is redundant and doesn't add to the story.

Reviewer #3 (Remarks to the Author):

Muller et al. report human semen inhibits ZIKV infections of cells from the anogenital region. Sexual transmission is an important topic of ZIKV research because the virus was reported to persist in male reproductive tracts. This reviewer has two major concerns. First, a mouse model for male-to-female transmission has been recently reported (Cell Rep. 2017 18(7):1751-1760. doi: 10.1016/j.celrep.2017.01.056). It was found that during mating, 73% of infected male mice transmitted ZIKV to uninfected females, and 50% of female mice became infected, with evidence of fetal infection in resulting pregnancies. How do the authors reconcile their results with the higher sexual transmission rate in the mouse model? Second, the current manuscript is quite descriptive, no mechanism was revealed at all.

Response to Reviewers

Reviewer #1 (Remarks to the Author):

The work of Müller et. al. is focused on the role of semen in Zika virus transmission. Sexual transmission of ZIKV is well established and therefore the importance of understanding mechanisms of this transmission and body defense systems are important. The authors used different experimental systems of Zika infection: cell lines primary cells from anogenital tract, explants of human cervix and proved that all of them support ZIKV replication. When they applied human semen, ZIKV replication in all these systems was inhibited. This was not due to the semen cytotoxicity because HIV replication in some of these systems was upregulated under semen treatment, confirming the authors' earlier observations. The authors conclude that semen contains protective factors against ZIKV and present some data that these factors affect early stages of ZIKV infection. Experiments are well-described, illustrated by convincing Figures and the data were subjected to appropriate statistical analysis.

We thank the reviewer for his/her positive comments.

I have the following critical remarks concerning the experiments and the way these experiments were interpreted:

1. Infection of ZIKV and HIV in TZM-bl cells were evaluated with different assays. While HIV infection was measured by fluorescence, the ZIKV infection was evaluated based on OD. How sensitive is the latter compared to the former. The authors claim that these cells were infectable by ZIKV. But could it be that the infection is so low and the assay is so insensitive that the stimulation can be missed?`

In our original submission, we used similar MOIs of HIV-1 and ZIKV for normalization. As the Reviewer points out, in our original submission baseline HIV-1 infection was assayed using a chemiluminescence-based reporter gene assay, and baseline ZIKV infection by quantifying OD of ZIKV E protein positive cells by the “in cell horse radish peroxidase based immuno assay”. To evaluate the effect of SEVI on both viruses using the same readout, we have now applied the same “in cell HRP immunodetection assay” for both viruses, by detecting ZIKV E protein for ZIKV, and the HIV-1 p24 protein for HIV. As now shown in Fig. S2b, using the same assay for the two viruses, infection rates ZIKV and HIV-1 were similar in TZM-bl cells. Under these conditions, SEVI did not affect ZIKV infection, whereas HIV-1 infection rates were enhanced by 11-fold, confirming our previous results.

2. The authors claim that “Amongst the tested cell lines, viral replication was least efficient in SKOV3 and T-84 cells, and most efficient in SW480, HeLa, and OVCAR-3 cells”. It is not clear how they compared the efficiency. Were the same number of cells infected? Was the cell density the same?

As now clarified in the methods section, all cells were seeded at 10,000 cells per well and infected with equal concentrations of virus. Thus, infection efficiencies between the cell lines could be directly compared.

3. The authors report about significant variability of the explants in terms of ZIKV production. Did the authors normalize by the weight or other methods since it is extremely difficult to make the tissue blocks similar in size?

Indeed, making tissue blocks of similar sizes is difficult, as we have learned over years of experience conducting *ex vivo* infections of human tissues (e.g. (Münch et al., Virology, 2005), (Münch et al., Cell, 2007a), (Münch et al., Cell, 2007b), (Schindler et al., Retrovirology, 2010), (Heigele et al., Retrovirology, 2014). Processing of tissues is performed by an experienced technical assistance. The size of the blocks is “normalized” by eye and all blocks that are larger or smaller than 2x1x1 mm are discarded. However, to minimize variability between samples, we infected individual tissue blocks (placed into a cavity of a 96 well plate) and then pooled six of the blocks into one gel-foam in one cavity of a 12 well plate (see Fig. 2), as previously reported by (Grivel and Margolis, Nat. Protoc., 2009) and (Arakelyan et al., Human Retroviruses, 2014). To directly determine the variability in ZIKV replication in the tissue blocks, we conducted an experiment where 48 blocks derived from vaginal tissue were infected and cultured individually. As shown in the new Fig. S1, ZIKV replicated in 32 out of 48 blocks with a virus load increase from median 2.2×10^4 to 8.0×10^5 . Thus, we feel that our conclusions that ZIKV is capable of replicating in these tissues is justified.

4. The description of the explants' culture should be made clearer. The authors should explain how “six blocks were incubated individually per 96 well” and how they are “transferred onto gelfoams in a 12 well plate”.

Please see revised Methods section (lines 447-456 and 543-548), where we have now clarified these details.

5. *Figure 3d. Did the authors normalize average by average or for every experiment?*

We had indeed normalized average by average for each experiment, but this figure has now been replaced by new Fig. S2b in which the effect of SEVI amyloid on ZIKV and HIV-1 infection was determined using the same target cells and assayed similarly (please see details above under point #1).

6. *P values should be added everywhere where the comparisons of means were performed.*

All results were re-analyzed by a statistician and P values are now given throughout the manuscript (please see methods section and figure legends).

Reviewer #2 (Remarks to the Author):

In this manuscript, the authors showed that cells from anogenital region are susceptible to ZIKV. Since previous papers showed that ZIKV can be sexually transmitted from male to female, they examined whether semen affects the virus infectivity in these cells. Interestingly, SEVI, which is known to boost HIV infection, did not change virus infectivity, but semen and seminal plasma were shown to inhibit ZIKV infection. The inhibitory effect of semen against ZIKV is interesting, however, there are a few key questions that should be asked in order to decipher the responsible mechanisms (e.g. identifying key molecules). Overall, the data are descriptive. Therefore, I think this manuscript is better suited for a more topic-specific journal.

Major points:

1. The authors should examine the mechanisms underlying why semen inhibits ZIKV infection. For example, they should try to identify which molecules (proteins, sugar, lipids, etc.) play an important role in the inhibition of ZIKV infection.

As suggested by this reviewer and also reviewer 3, we have now further examined the mechanism underlying the anti-ZIKV effects of semen. Our previous results demonstrated that the antiviral effect is reversible, mediated by a soluble factor, occurs during attachment/entry, and is donor-independent. We now have conducted extensive additional experiments to decipher which molecules (proteins, sugar, lipids, etc) play an important role as suggested by the reviewer, in addition to new experiments characterizing the mechanistic basis of inhibition in other ways. In summary, since the original submission, we have demonstrated that:

- SP blocks the attachment of Zika virions to target cells (Fig. 5b and S8c-d)
- SP also inhibits infection of Dengue and West Nile virus (Fig. 7b and 7c and S12a) whilst enhancing HIV-1 infection (Fig. 7 d and 7e); thus SP seems to exert pan-anti-flavivirus activity
- Proteinase K (Fig. S9a), boiling (Fig S9b), acid treatment (Fig. S9c), or removal of glycoproteins (Fig. S9d) does not abrogate the anti-ZIKV activity of SP, suggesting that the responsible factor is not a polypeptide or glycoprotein
- the antiviral factor has a molecular weight larger than 300 kDa but is less than ~ 200 nm in diameter (Fig. 6a, b)
- a SP fraction containing extracellular vesicles (EVs) inhibit ZIKV as efficiently as unprocessed SP (Fig. 6c) and boiling this EV fraction does not abrogate antiviral activity (Fig 6d)
- SP-derived EVs block attachment of Zika virions to target cells (Fig. 6e and Fig S10), similar as unprocessed SP (Fig. 5b and S8c-d).

These novel findings shed light on the mechanism of ZIKV inhibition by SP, and show that EVs in semen prevent ZIKV attachment to target cells. EVs are abundantly present in semen and have been shown to play roles in sperm motility, semen liquefaction, and antimicrobial defense, an aspect which is now discussed (lines 313-328).

2. Fig.3c and d. They claim that SEVI did not enhance ZIKV infection in vitro. However, it seems that GWUH infectivity is enhanced after 50ug/ml SEVI treatment compared with 0 ug/ml SEVI treatment.

The reviewer is correct that there was a slight (but non-significant) increase of ZIKV GWUH infection in the presence of 50 µg/ml SEVI on TZM-bl cells. We have repeated these experiments several times and now show a more representative experiment where ZIKV MR766, ZIKV GWUH, and HIV-1 were analyzed in parallel. Fig S2a (Vero E6 target cells) and Fig S2b (TZM-bl target cells) demonstrate that ZIKV infection rates are in fact not significantly enhanced by SEVI.

3. *They showed that primary vaginal and endometrial tissue is susceptible to ZIKV (Fig. 2). So, testing the inhibitory effect of semen against ZIKV (related to Fig. 9) in these primary tissue cells would strengthen their conclusion.*

We thank the reviewer for this helpful comment and conducted the requested experiment. Due to the high variability in virus replication in the tissues (please see also comments to reviewer 1 and the new Fig. S1), we increased sample size to reduce variability and to increase statistical power. For this, we prepared 2 x 24 vaginal tissue blocks from donor A, and 2 x 48 tissue blocks of donor B. The blocks were incubated with ZIKV GWUH in presence of 25% semen or 25% PBS. After washing, blocks were transferred into separate 96 wells and incubated at 37°C. Aliquots were taken at days 0 (wash control), and day 4 or 5 and examined for viral genome copy numbers by qPCR. ZIKV replicated in tissues of both donors (Fig. 9a and 9b). In the presence of semen, virus loads were reduced on average by 79 % (P=0.0198) (Fig. 9a) and 74 % (P=0.0326) (Fig. 9b), respectively. A cytotoxicity assay performed with tissue blocks of donor A revealed no signs of reduced viability in the presence of SE (Fig. 9c), excluding the possibility that cytotoxicity was accounting for the observed effects. These results demonstrate that SE also suppresses ZIKV replication in *ex vivo* infected vaginal tissues.

4. *Although few case of sexual transmission of flavivirus other than ZIKV have been reported, it would be informative to test whether semen also suppresses other flaviviruses such as dengue virus.*

We followed this great suggestion and now show that semen inhibits Dengue virus and West Nile virus (both flaviviruses), suggesting that semen may exert pan-anti-flavivirus activity, potentially explaining why members of this virus family are not efficiently transmitted via sexual intercourse (Fig. 7b, 7c, S12a and lines 326-328 and 349-355 in the discussion).

Specific points:

5. *They should show more thorough statistical analysis on their data. Without proper statistical analysis, it is difficult to decipher the impact of some of their experiments.*

All results were re-analyzed by a statistician and P values are now given throughout the manuscript (please see methods section and figure legends).

6. *Fig. 2. They only showed the highest detected virus titers, but they should show all data related to the replication of ZIKV in cells isolated from VT and ET.*

We now show the requested data in the new Fig. 2.

7. *Why are there no error bars in Fig. 4 c-e? Have they done only one experiment?*

In the previous figure, we had shown data obtained from one representative experiment. We now show average data obtained from three independent experiments ± SD (Fig. 3).

8. *The experiment in Fig.5 is confusing. Usually CPE is not visible when virus cannot replicate/spread efficiently in the cells. It seems that this data is redundant and doesn't add to the story.*

Given that this data is redundant with other presented experiments, we have moved the figure to the supplement (Fig. S6).

Reviewer #3 (Remarks to the Author):

Muller et al. report human semen inhibits ZIKV infections of cells from the anogenital region. Sexual transmission is an important topic of ZIKV research because the virus was reported to persist in male reproductive tracts. This reviewer has two major concerns. First, a mouse model for male-to-female transmission has been recently reported (Cell Rep. 2017 18(7):1751-1760. doi: 10.1016/j.celrep.2017.01.056). It was found that during mating, 73% of infected male mice transmitted ZIKV to uninfected females, and 50% of female mice became infected, with evidence of fetal infection in resulting pregnancies.

How do the authors reconcile their results with the higher sexual transmission rate in the mouse model?

Mouse models have certainly advanced our understanding of Zika virus transmission and pathogenicity, but come with the caveat that these models rely on immunocompromised (interferon- α/β and $-\gamma$ receptor knockout) animals. As such, the models have their limits, and in fact, the authors of the aforementioned study concluded in their discussion “the relevance of these studies to human infection with ZIKV is unknown”. In addition, the reported 73 % transmission rates during mating in mice do not reflect sexual ZIKV transmission rates in humans where cases of sexual transmission are still rare. Thus, our finding that human semen inhibits ZIKV infection may provide an explanation for the seemingly low sexual transmission rates of ZIKV. Whether murine semen also inhibits remains to be determined.

Second, the current manuscript is quite descriptive, no mechanism was revealed at all.

We kindly refer to our response to reviewer 2 in which we summarize our new findings on the mechanism. Our new data demonstrate that extracellular vesicles in semen inhibit infection by preventing Zika virus attachment to target cells.

Reviewers' comments:

Reviewer #1 (Remarks to the Author):

The authors revised their paper and added significant amount of new data.

Currently, the results of the study can be summarized in three points: (i) ZIKV infects cells and explants of the female lower genital tract; (ii) this effect is inhibited by semen or seminal plasma; and (iii) this effect is mediated by extracellular vesicles (EVs).

Critical remarks.

I. It had been established earlier that ZIKV has a broad cellular tropism including cells of the anogenital tract. The results reported in this paper confirm this observation. Also, it had earlier been found that ZIKV replicates in the non-human primate vagina upon intravaginal inoculation. Finally, in clinical studies it was shown that ZIKV RNA is found in the human vagina. Thus, the fact that ZIKV replicates in vaginal and endometrial explants is expected, although it useful to have this shown in the explant system.

II. The finding that semen suppresses ZIKV infection in these systems is new, to the best of my knowledge, and is relevant to transmission in vivo.

III. The evidence presented that extracellular vesicles mediate this effect is not convincing and requires more proof.

Centrifugation may result not only in sedimentation of EVs but also in large protein aggregates. The authors should characterize this fraction more thoroughly. Filtering through filters with different pore sizes is not enough. The authors should use NTA to characterize the size distribution of their fraction. Also, to demonstrate the specificity of the EV effects, the authors should use EVs of similar size from another source. Finally, it is striking that boiling this preparation, which should destroy these lipid vesicles, does not inhibit the activity of the fractions. As I see it, this is evidence against EV involvement in the inhibitory effect.

Minor critical remarks

A. Experiments with Dengue and West Nile viruses, which seem also to be inhibited by semen, are interesting but contrast more thorough and detailed experiments with ZIKV. It is important to repeat with them all the experiments performed with ZIKV virus in order to justify conclusions about the generality of the phenomenon of inhibition of infection of flaviviruses by semen. Alternatively, these experiments could constitute a separate paper.

B. The authors statement that small vesicles are called exosomes (line 234) is incorrect. Exosomes are vesicles that are generated by cells through multivesicular bodies and though differing in origin may be of the same size as small vesicles that bud from the plasma membrane.

C. The authors should provide more details on the sources of the tissues they used. At one point (line 122) the authors claim the tissues they used were biopsies from the vagina. However, in Methods (line 384) it is said that tissues come from hysterectomy samples. The latter presumably consist predominantly of uterus and cervix.

D. The amount of semen needed to block tissue infection is much higher than to inhibit infection of isolated cells: (25% semen for 73% inhibition of infection vs.1% of semen to inhibit by 90%). The authors need to comment this difference.

Reviewer #2 (Remarks to the Author):

Remarks to authors

The authors have performed additional experiments to respond to the reviewer's comments and this revised manuscript is improved. However, I still have a few concerns about the data they provide in this revised manuscript.

1: When they performed the ZIKV virion attachment assay, what temperature did they use? The attachment assay is usually performed at 4 C, and if performed at 37 C virus is able to enter cells and thus the experiment becomes an entry assay. In the manuscript, the authors should distinguish 'entry' and 'attachment'. It appears that their IF data represents not only 'virion attachment' but the 'virion invasion' step. As they are presented, the data and designed experiments are not sufficient to examine the inhibitory effect of SP on attachment.

2: Fig. 5, were cells washed after the treatment with SP before adding ZIKV?

3: Fig. 6, the authors conclude that EV in SP blocks virus attachment. As mentioned at the point 1, this experiment is not sufficient to examine the inhibitory effect of SP on attachment. While this observation is, in my opinion, sufficient for this manuscript, it will be interesting to further pursue the underlying mechanism.

Reviewer #3 (Remarks to the Author):

The authors have adequately addressed the points of this reviewer.

Reviewer #1 (Remarks to the Author):

“The authors revised their paper and added significant amount of new data. Currently, the results of the study can be summarized in three points: (i) ZIKV infects cells and explants of the female lower genital tract; (ii) this effect is inhibited by semen or seminal plasma; and (iii) this effect is mediated by extracellular vesicles (EVs).

Critical remarks.

I. It had been established earlier that ZIKV has a broad cellular tropism including cells of of the anogenital tract. The results reported in this paper confirm this observation. Also, it had earlier been found that ZIKV replicates in the non-human primate vagina upon intravaginal inoculation. Finally, in clinical studies it was shown that ZIKV RNA is found in the human vagina. Thus, the fact that ZIKV replicates in vaginal and endometrial explants is expected, although it useful to have this shown in the explant system.

II. The finding that semen suppresses ZIKV infection in these systems is new, to the best of my knowledge, and is relevant to transmission in vivo.”

We thank the reviewer for acknowledging our findings.

“III. The evidence presented that extracellular vesicles mediate this effect is not convincing and requires more proof. Centrifugation may result not only in sedimentation of EVs but also in large protein aggregates. The authors should characterize this fraction more thoroughly. Filtering through filters with different pore sizes is not enough. The authors should use NTA to characterize the size distribution of their fraction.”

As suggested, we now performed NTA (lines 546-549). We show that the EV preparation from SP that inhibits ZIKV infection contains vesicles with a diameter of 141.6 ± 3.7 nm (new Fig. S11a and text lines 222-224), which is in agreement with published size of EVs in semen (Ronquist G et al., Biochim Biophys Acta, 1985; Yang C et al., Andrology, 2017).

“Also, to demonstrate the specificity of the EV effects, the authors should use EVs of similar size from another source.”

As suggested, we prepared EVs from urine and saliva and characterized the samples using NTA. The EV samples from urine and saliva contained vesicles with similar or slightly increased average size as compared to those of SP. However, in contrast to EVs from SP, EVs from urine and saliva did not affect ZIKV infection (new Fig. S13 and text lines 227-228).

“Finally, it is striking that boiling this preparation, which should destroy these lipid vesicles, does not inhibit the activity of the fractions. As I see it, this is evidence against EV involvement in the inhibitory effect.”

EVs from SP are rich in sphingomyelin and cholesterol and display significant rigidity and stability (reviewed by Yáñez-Mó M et al., J Extracell Vesicles, 2015 and Kooijmans S et al., Int J Nanomedicine, 2012). In fact, when boiling the SP EV sample, NTA revealed that the size distribution of the vesicles was only slightly altered (new Fig. S11c), explaining why boiled EVs from SP (Fig. 6d) and boiled SP (Fig. S10b) retain antiviral activity.

“Minor critical remarks

A. Experiments with Dengue and West Nile viruses, which seem also to be inhibited by semen, are interesting but contrast more thorough and detailed experiments with ZIKV. It is important to repeat with them all the experiments performed with ZIKV virus in order to justify conclusions about the generality of the phenomenon of inhibition of infection of flaviviruses by semen. Alternatively, these experiments could constitute a separate paper.”

Experiments with Dengue and West Nile viruses were performed upon request by another reviewer. However, we have now cautioned our discussion on the generalization of this phenomenon and state that additional experiments are required to clarify whether SE has a broad-based anti-flavivirus activity (lines 352-354)

“B. The authors statement that small vesicles are called exosomes (line 234) is incorrect. Exosomes are vesicles that are generated by cells through multivesicular bodies and though differing in origin may be of the same size as small vesicles that bud from the plasma membrane.”

We thank the reviewer and corrected (lines 220-221).

C. The authors should provide more details on the sources of the tissues they used. At one point (line 122) the authors claim the tissues they used were biopsies from the vagina. However, in Methods (line 384) it is said that tissues come from hysterectomy samples. The latter presumably consist predominantly of uterus and cervix.

We apologize for not making this clearer. Tissues were derived from pelvic organ prolapse patients that underwent surgery where cervical and vaginal tissue was removed. This has now been clarified (lines 378-382).

D. The amount of semen needed to block tissue infection is much higher than to inhibit infection of isolated cells: (25% semen for 73% inhibition of infection vs. 1% of semen to inhibit by 90%). The authors need to comment this difference.

Tissue infection was measured by determining progeny virus released from infected tissue after 4-5 days which differs from directly measuring primary infection of cells by antibody staining. In combination with the high viral inoculums used to achieve reliable infection of the tissues and viral spread over time overall observed effects are more moderate. We have now commented on this (lines 273-275).

Reviewer #2 (Remarks to the Author):

“Remarks to authors

The authors have performed additional experiments to respond to the reviewer’s comments and this revised manuscript is improved. However, I still have a few concerns about the data they provide in this revised manuscript.

1: When they performed the ZIKV virion attachment assay, what temperature did they use? The attachment assay is usually performed at 4 C, and if performed at 37 C virus is able to enter cells and thus

the experiment becomes an entry assay. In the manuscript, the authors should distinguish 'entry' and 'attachment'. It appears that their IF data represents not only 'virion attachment' but the 'virion invasion' step. As they are presented, the data and designed experiments are not sufficient to examine the inhibitory effect of SP on attachment."

We have performed the attachment experiments at 4°C and 37°C. Results of both experimental conditions are now shown in the revised manuscript and confirm that semen blocks ZIKV attachment (Fig. 5b,c, Fig. S8 and S9).

2: Fig. 5, were cells washed after the treatment with SP before adding ZIKV?

Yes, cells were washed in between. This is stated in lines 195-200 and the figure legend.

3: Fig. 6, the authors conclude that EV in SP blocks virus attachment. As mentioned at the point 1, this experiment is not sufficient to examine the inhibitory effect of SP on attachment. While this observation is, in my opinion, sufficient for this manuscript, it will be interesting to further pursue the underlying mechanism.

As mentioned above for SP (1.) we have also confirmed that SP derived EVs block virion attachment at 4°C and 37°C (Fig. 6 e,f, Fig S12).

Reviewer #3 (Remarks to the Author):

The authors have adequately addressed the points of this reviewer.

Reviewers' comments:

Reviewer #1 (Remarks to the Author):

The authors significantly improved the manuscript.

However, one point still bothers me: The fact that boiling of the EV preparation does not destroy its activity. In their response the authors refer to the reviews that refer to papers showing the stability of EVs. However, in these papers the authors demonstrated that freezing-thawing does not destroy EVs. Boiling denatures the proteins, oxidizes unsaturated phospholipids and, in my mind, destroys the structure. The fact that the activity remains is an evidence against the role of EVs.

The remnants of EVs may aggregate after boiling and give rise to the particles that the authors register with NTA. I think that the authors should present additional analysis of the boiled preparation, e.g., EM images.

Reviewer #2 (Remarks to the Author):

The authors have addressed all points of this reviewer.

Reviewer #1 (Remarks to the Author):

The authors significantly improved the manuscript.

However, one point still bothers me: The fact that boiling of the EV preparation does not destroy its activity. In their response the authors refer to the reviews that refer to papers showing the stability of EVs. However, in these papers the authors demonstrated that freezing-thawing does not destroy EVs. Boiling denatures the proteins, oxidizes unsaturated phospholipids and, in my mind, destroys the structure. The fact that the activity remains is an evidence against the role of EVs. The remnants of EVs may aggregate after boiling and give rise to the particles that the authors register with NTA. I think that the authors should present additional analysis of the boiled preparation, e.g., EM images.

We were also bothered by the fact that boiling does not destroy anti-ZIKV activity of EVs. However, as shown by NTA in the previous version of the paper, boiling only moderately changed size distribution of the EVs (Fig. S11a-c). As suggested, we now also performed EM analysis of untreated and boiled SP-derived EVs. As shown in new Fig. S11d and described in lines 224-226 (M&M lines 557-562), also boiling only marginally changed the morphology of the vesicles. These data are also in agreement with reported data showing that vesicles are relatively stable structures and survive 100°C (Mansy, S.S., and Szostak, J.W. ,Proc. Natl. Acad. Sci., 2008; reviewed by Yáñez-Mó M et al., J Extracell Vesicles, 2015 and Kooijmans S et al., Int J Nanomedicine, 2012). In summary, we are convinced that these new imaging data together with our previous findings obtained by NTA provide compelling evidence that EVs are the components in semen that are responsible for the anti-ZIKV activity.

Reviewer #2 (Remarks to the Author):

The authors have addressed all points of this reviewer.

REVIEWERS' COMMENTS:

Reviewer #1 (Remarks to the Author):

In my previous remarks, I was skeptical that boiling of extracellular vesicles (EVs) fails to destroy their activity against ZIKV. Here, the authors show that both before and after boiling EVs reveal similar structures in their preparations. Also, the authors refer to papers in support of their claim of EV resistance to boiling. However, two of the three papers they referred to are reviews, and the only original paper describes the heat resistance of artificial "simple protocells" composed of simple single-chain amphiphiles such as fatty acids, fatty alcohols, and fatty-acid glycerol esters. With regard to the EM presented by the authors, I agree that similar structures can be found in both boiled and initial preparations. However, I am not sure that these are extracellular vesicles as defined by the Society of Extracellular Vesicles that recommend the use of 3 or more EV markers with at least one being a tetraspanin protein (see Lotvall et al., J Extracell Vesicles 2014). Anyway something in their preparation withstands boiling and remains active.

I suggest that the authors make an additional experiment: while they showed that their EVs are not destroyed by boiling and preserve their anti-ZIKV activity, a negative control is absent. The authors should destroy their EVs by sonication and show that in this case the effect disappears. Then their hypothesis on EVs seems to be more convincing. Except for the paragraph on EVs, the paper seems satisfactory to me.

Comments by the Reviewer:

Reviewer #1 (Remarks to the Author):

“In my previous remarks, I was skeptical that boiling of extracellular vesicles (EVs) fails to destroy their activity against ZIKV. Here, the authors show that both before and after boiling EVs reveal similar structures in their preparations. Also, the authors refer to papers in support of their claim of EV resistance to boiling. However, two of the three papers they referred to are reviews, and the only original paper describes the heat resistance of artificial “simple protocells” composed of simple single-chain amphiphiles such as fatty acids, fatty alcohols, and fatty-acid glycerol esters. With regard to the EM presented by the authors, I agree that similar structures can be found in both boiled and initial preparations. However, I am not sure that these are extracellular vesicles as defined by the Society of Extracellular Vesicles that recommend the use of 3 or more EV markers with at least one being a tetraspanin protein (see Lotvall et al., J Extracell Vesicles 2014). Anyway something in their preparation withstands boiling and remains active.

I suggest that the authors make an additional experiment: while they showed that their EVs are not destroyed by boiling and preserve their anti-ZIKV activity, a negative control is absent. The authors should destroy their EVs by sonication and show that in this case the effect disappears. Then their hypothesis on EVs seems to be more convincing. Except for the paragraph on EVs, the paper seems satisfactory to me.”

We acknowledge the reviewers comments. The exact nature of the structures in the EV preparation from semen that are responsible for the antiviral activity will be determined in future work. We have therefore toned down our conclusions and stated this in the according paragraphs in the results and discussion section (lines 195-220 and 298-318).